# ADAFOCAL:
# CALIBRATION-AWARE ADAPTIVE FOCAL LOSS

## ABSTRACT

Much recent work has been devoted to the problem of ensuring that a neural network's confidence scores match the true probability of being correct, i.e. the calibration problem. Of note, it was found that training with focal loss leads to better calibrated deep networks than cross-entropy, while achieving the same level of accuracy Mukhoti et al. (2020). This success stems from focal loss regularizing the entropy of the network's prediction (controlled by the hyper-parameter $\gamma$), thereby reining in the network's overconfidence. Further improvement is expected if $\gamma$ is selected independently for each training sample. However, the proposed Sample-Dependent Focal Loss (FLSD) in Mukhoti et al. (2020) is based on simple heuristics that does not take into account the difference in the network's calibration behaviour for different samples (or groups of samples). As a result it is only slightly better than focal loss with fixed $\gamma$. In this paper, we propose a calibration-aware version of FLSD, called AdaFocal, which, at every training step $t$, adaptively modifies the $\gamma$ for individual group of samples based on (1) $\gamma_{t-1}$ from the previous training step (2) the magnitude of the network's under/over-confidence for those groups. We evaluate our method on various small to large-scale image recognition tasks and one NLP task, covering a variety of network architectures, to confirm that AdaFocal consistently achieves improved calibration without a significant loss in accuracy. Further, the models trained with AdaFocal are shown to have significantly improved Out-of-Distribution (OOD) detection capability.

## 1 INTRODUCTION

Neural networks have found tremendous success in almost every field including computer vision, natural language processing, and speech recognition. Over time, these networks have grown complex and larger in size to achieve state-of-the-art performance and they continue to evolve further in that direction. However, it has been well established that such high capacity networks suffer from poor calibration Guo et al. (2017), i.e. the confidence scores of the predictions do not reflect the real world probabilities of those predictions being true. For example, if the network assigns $0.8$ confidence to a set of predictions, we should expect $80\%$ of those predictions to be correct. However, this is far from reality since modern networks tend to be grossly over-confident. This is of great concern, particularly for mission-critical applications such as autonomous driving, medical diagnosis, wherein the downstream decision making not only rely on the predictions but also on their confidence.

In recent years, there has been a growing interest in developing methods for calibrating neural networks. These can be mainly divided into two categories (1) post-hoc approaches that perform calibration after training (2) methods that calibrate the model during training itself. The first includes methods such as Platt scaling Platt (1999), histogram binning Zadrozny & Elkan (2001), Isotonic regression Zadrozny & Elkan (2002), Bayesian binning and averaging Naeini et al. (2015); Naeini & Cooper (2016), and Spline fitting Gupta et al. (2021). Methods in the second category focus on training the model on an objective function that accounts for calibration as well, including Maximum Mean Calibration Error (MMCE) Kumar et al. (2018), Label smoothing Müller et al. (2019), and recently focal loss Mukhoti et al. (2020). These methods aim to produce inherently calibrated models which when combined with post training calibration methods lead to further improvements.

**Contribution.** Our work falls into the second category. We build upon the calibration properties of focal loss to propose a modification that further improves its performance. Firstly, we make the observation that while regular focal loss, with a fixed $\gamma$ parameter, improves the overall calibration by preventing samples from being over-confident, it also leaves other samples under-confident. To

address this drawback, we propose a modification to the focal loss called AdaFocal that adjusts the $\gamma$ for each training sample (or rather a group of samples) separately by taking into account the model's under/over-confidence about a similar corresponding group in the validation set. We evaluate the performance of our method on four image classification tasks: CIFAR-10, CIFAR-100, Tiny-ImageNet and ImageNet, and one text classification task: 20 Newsgroup, using various model architectures, and show that AdaFocal substantially outperforms the regular focal loss and other state-of-the-art calibration techniques in the literature. We further study the performance of AdaFocal on an out-of-distribution detection task and find it to perform better than the competing methods. Finally, we find that the models trained using AdaFocal get innately calibrated to a level that most times do not significantly benefit from temperature scaling.

## 2 PROBLEM SETUP AND DEFINITIONS

Consider a classification setting where we are given a set of training data $\{(\mathbf{x}_n, y_{\text{true},n})\}$, with $\mathbf{x}_n \in \mathcal{X}$ being the input and $y_{\text{true},i} \in \mathcal{Y} = \{1, 2, \ldots, K\}$ the associated ground-truth label. Using this data we wish to train a classifier $f_\theta(\mathbf{x})$ that outputs a vector $\hat{\mathbf{p}}$ over the $K$ classes. We also assume access to a validation set for hyper-parameter tuning and a test set for evaluating its performance. For example, $f_\theta(\cdot)$ can be a neural network with learnable parameters $\theta$, $\mathbf{x}$ is an image, and $\hat{\mathbf{p}}$ is the output of a *softmax layer* whose $k^{\text{th}}$ element $\hat{p}_k$ is the probability score for class $k$. We refer to $\hat{y} = \arg\max_{k \in \mathcal{Y}} \hat{p}_k$ as the network's prediction and the associated probability score $\hat{p}_{\hat{y}}$ as the predicted confidence, and the same quantity for the $j$th example is $\hat{p}_{\hat{y},j}$.

In this setting, a network is said to be perfectly calibrated if the predicted confidence $\hat{p}_{\hat{y}}$ reflects the true probability of the network classifying $\mathbf{x}$ correctly i.e. $\mathbb{P}(\hat{y} = y_{\text{true}} \mid \hat{p}_{\hat{y}} = p) = p, \ \forall p \in [0, 1]$ Guo et al. (2017). Continuing our example, if the network assigns an average confidence score of $0.8$ to a set of predictions then we should expect $80\%$ of those to be correct. We define *Calibration Error* as $\mathcal{E} = \hat{p}_{\hat{y}} - \mathbb{P}(\hat{y} = y_{\text{true}} \mid \hat{p}_{\hat{y}})$ and the *Expected Calibration Error* as $\mathbb{E}_{\hat{p}_{\hat{y}}}[\mathcal{E}] = \mathbb{E}_{\hat{p}_{\hat{y}}}[\,|\hat{p}_{\hat{y}} - \mathbb{P}(\hat{y} = y_{\text{true}} \mid \hat{p}_{\hat{y}})|\,]$ Guo et al. (2017). However, as the true calibration error cannot be computed empirically with a finite sized dataset, the following three approximations are generally used in the literature. That is, for a dataset $\{(\mathbf{x}_n, y_{\text{true},n})\}_{n=1}^N$, (1) $\text{ECE} = \sum_{i=1}^M \frac{|B_i|}{N} |C_i - A_i|$ Guo et al. (2017), where $B_i$ is equal-width bin that contains all examples $j$ with $\hat{p}_{\hat{y},j}$ in the range $[\frac{i}{M}, \frac{i+1}{M})$, $C_i = \frac{1}{|B_i|} \sum_{j \in B_i} \hat{p}_{\hat{y},j}$ is the average confidence and $A_i = \frac{1}{|B_i|} \sum_{j \in B_i} \mathbb{1}(\hat{y}_j = y_{\text{true},j})$ is the bin accuracy. Note that $E_i = C_i - A_i$ is the empirical approximation of the calibration error $\mathcal{E}$, (2) $\text{AdaECE} = \sum_{i=1}^M \frac{|B_i|}{N} |C_i - A_i|$ Nguyen & O'Connor (2015), where $\forall i, j \ |B_i| = |B_j|$ are adaptively sized (equal-mass) bins that contain an equal number of samples, and (3) ClasswiseECE Kumar et al. (2018); Kull et al. (2019) estimates the calibration over all $K$ classes: $\text{ClasswiseECE} = \frac{1}{K} \sum_{i=1}^M \sum_{k=1}^K \frac{|B_{i,k}|}{N} |C_{i,k} - A_{i,k}|$ where $C_{i,k} = \frac{1}{|B_{i,k}|} \sum_{j \in B_{i,k}} \hat{p}_{k,j}$ is the average confidence for the $k$th class and $A_{i,k} = \frac{1}{|B_{i,k}|} \sum_{j \in B_{i,k}} \mathbb{1}(y_{\text{true},j} = k)$ is the accuracy of the $k$th class in the $i$th bin.

Lastly, as ECE has been shown to be a biased estimate of true calibration Vaicenavicius et al. (2019), we additionally use two de-biased estimates of ECE namely $\text{ECE}_{\text{debiased}}$ proposed in Kumar et al. (2019) and $\text{ECE}_{\text{sweep}}$ proposed in Roelofs et al. (2021) to further confirm our results.

## 3 CALIBRATION PROPERTIES OF FOCAL LOSS

Focal loss Lin et al. (2017) $\mathcal{L}_{FL}(p) = -(1 - p)^\gamma \log p$ was originally proposed to improve the accuracy of classifiers by focusing on hard examples and down-weighting well classified examples. Recently it was further shown that focal loss may also result in significantly better calibrated models than cross entropy Mukhoti et al. (2020). This is because, based on the relation: $\mathcal{L}_{FL} \geq KL(q||\hat{\mathbf{p}}) - \gamma\mathbb{H}(\hat{\mathbf{p}})$ where $q$ is the one-hot target vector, focal loss while minimising the main KL divergence objective also increases the entropy of the prediction $\hat{\mathbf{p}}$. As a consequence this prevents the network from being overly confident on wrong predictions and overall improves calibration.

The regular focal loss with fixed $\gamma$, as we show in this section, does not achieve the best calibration. In Figure 1, we plot the calibration behaviour of ResNet50 in different bins when trained on CIFAR-10 with different focal losses. The $i$th bin's calibration error subscripted by "val" $E_{val,i} = C_{val,i} - A_{val,i}$ is computed on the validation set using $15$ equal-mass binning. The figure shows the lowest (bin-0), a middle (bin-7) and highest bin (bin-7). For reference, the rest of the bins and their bin boundaries are

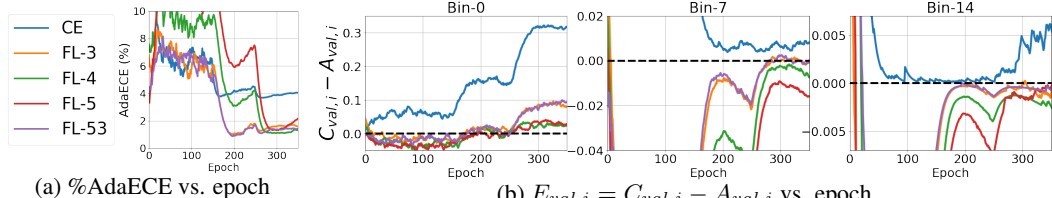

(a) %AdaECE vs. epoch                    (b) $E_{val,i} = C_{val,i} - A_{val,i}$ vs. epoch

Figure 1: Calibration behaviour of ResNet-50 trained on CIFAR-10 with cross entropy (CE), focal loss $\gamma = 3, 4, 5$ (FL-3/4/5) and FLSD-53 (or FL-53). The statistics are computed using 15 equal-mass binning on the validation set. (a) AdaECE, and (b) Calibration error $E_{val,i} = C_{val,i} - A_{val,i}$ for a lower (bin-0), middle (bin-7), and upper (bin-14) bin. The black horizontal lines in (b) represent $E_{val,i} = 0$. These exemplify the downside of regular focal loss i.e. although FL-4 achieves the overall lowest calibration error (AdaECE), the best performing $\gamma$ is different for different bins.

shown in Appendix B. From Figure 1 (a), we see that although focal loss $\gamma = 4$ achieves the overall lowest calibration error (AdaECE), there's no single $\gamma$ that performs the best across all the bins. For example, in bin-0 $\gamma = 4, 5$ seems to achieve better calibration whereas $\gamma = 0, 3$ are over-confident. For bin-7, on the other hand, $\gamma = 3$ seems to be better calibrated whereas $\gamma = 4, 5$ are under-confident and $\gamma = 0$ is over-confident.

This clearly indicates that using different $\gamma$s for different bins can further improve the calibration. Such an attempt is presented in Mukhoti et al. (2020) called the Sample-Dependent Focal Loss (FLSD-53) which assigns $\gamma = 5$ if the training sample's true class posterior $\hat{p}_{y_{\text{true}}} \in [0, 0.2)$ and $\gamma = 3$ if $\hat{p}_{y_{\text{true}}} \in [0.2, 1]$. However, this strategy is fixed for every dataset-model pair and is based on simple heuristics of choosing higher $\gamma$ for smaller values of $\hat{p}_{y_{\text{true}}}$ and relatively lower $\gamma$ for higher values of $\hat{p}_{y_{\text{true}}}$. However, from Figure 1(b), we see that FLSD-53 is also not the best strategy across all the bins. This, therefore, motivates the design of a $\gamma$ selection strategy that can assign an appropriate $\gamma$ for each bin based on the magnitude and sign of $E_{val,i}$. However, in order to design such a strategy we need solutions to the following two major challenges:

1. How do we find some correspondence between the "confidence of training samples", which we can manipulate during training by adjusting the entropy regularising parameter $\gamma$, and the "confidence of the validation samples", which we want to be actually manipulated but do not have direct control over? In other words, in order to indirectly control the confidence of a particular group of validation samples, how do we know which particular group of training samples' confidence to be manipulated?

2. Given that there is a correspondence between a training group and a validation group (even if it's loose), how do we arrive at the exact values of $\gamma$ that will lead to better calibration?

We try to answer the first question in the next section and the answer to the second question leads to AdaFocal which is the main contribution of the paper.

## 4    CORRESPONDENCE BETWEEN CONFIDENCE OF TRAIN AND VAL. SAMPLES

In order to find some correspondence, an intuitive thing to do would be to group the validation samples into $M$ equal-mass validation-bins, and then use these validation-bin boundaries to group the training samples as well. Then, we can compare the average confidence of the validation samples and the average confidence of the training samples, in the same validation-bin, to check for any correspondence.

**Quantities of interest**    For binning validation samples, we always look at the confidence of the top predicted class $\hat{y}$ denoted by $\hat{p}_{val,top}$ (bin average: $C_{val,top}$). For training samples, on the other hand, instead of the confidence of the top predicted class $\hat{y}$ denoted by $\hat{p}_{train,top}$ (bin average: $C_{train,top}$), we will focus on the confidence of the true class $y_{\text{true}}$ denoted by $\hat{p}_{train,true}$ (average: $C_{train,true}$) because during training we only care about $\hat{p}_{train,true}$ which is manipulated through some loss function. For reference however, Figure 10 in Appendix C compares $C_{train,true}$ and $C_{train,top}$ to show that as the training set accuracy approaches $100\%$, the top predicted class and the true class for a training sample become the same. Henceforth, for a cleaner notation, we will always refer to $C_{train} \equiv C_{train,true}$ and $C_{val} \equiv C_{val,top}$.

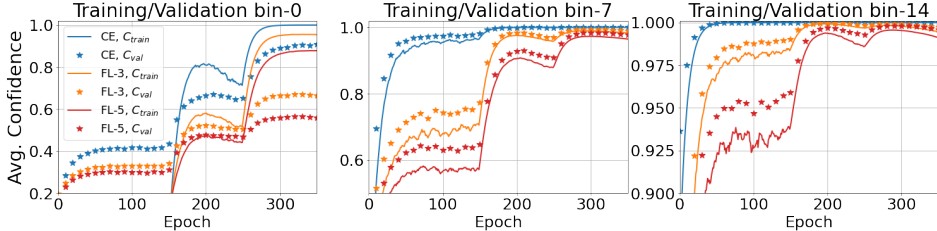

(a) **Independent binning**: training samples and validation samples are grouped independently into training-bins and validation-bins respectively. **Solid** line: $C_{train}$, and **Starred line**: $C_{val}$.

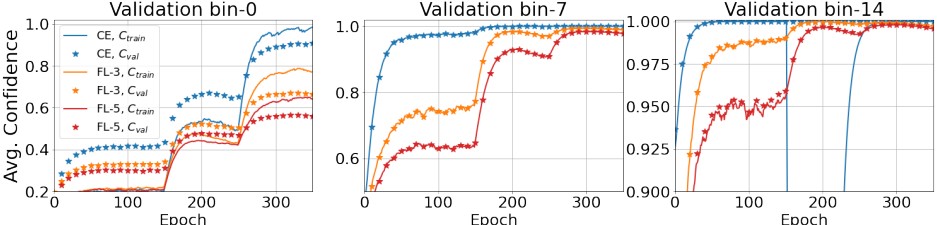

(b) **Common binning**: training samples grouped using validation-bin boundaries. **Solid**: $C_{train}$, **Starred** $C_{val}$

Figure 2: Correspondence between average confidence of a group of training samples ($C_{train}$) and a group of validation samples ($C_{val}$) for ResNet-50 trained on CIFAR-10 with focal loss $\gamma = 0$ (CE), $3, 5$. The binning involves 15 equal-mass bins with a lower (bin-0), middle (bin-7) and upper (bin-14) shown here. We see a good correspondence between $C_{train}$ (solid lines) and $C_{val}$ (starred lines).

**Common binning**    When training samples are grouped using the bin boundaries of the validation-bins. In Figure 2(b), we compare $C_{train,i}$ in validation-bin-$i$ [1] with $C_{val,i}$ in the same validation-bin-$i$, and find that there is indeed a good correspondence between the two quantities. For example in Figure 2(b), as $\gamma$ increases from 0, 3 to 5, the solid-line ($C_{train,i}$) gets lower, and the same behaviour is observed on the starred-line ($C_{val,i}$) as well. For completeness, rest of the bins are shown in Figure 12 Appendix C. This is very encouraging as now we can expect (even though loosely) that if we increase/decrease the confidence of a group of training samples in some lower (or middle, or higher) probability region then the same will be reflected on a similar group of validation samples in lower (or middle, or higher) probability region. This therefore provides a way to indirectly control the value of $C_{val,i}$ by manipulating $C_{train,i}$, and from a calibration point of view, our strategy going forward would be to exploit this correspondence to keep $C_{train,i}$ (which we have control over during training) closer to $A_{val,i}$ (the validation set accuracy in validation-bin-$i$) so that $C_{val,i}$ also stays closer to $A_{val,i}$ to overall reduce the calibration error $E_{val,i} = C_{val,i} - A_{val,i}$.

**Independent binning**    Before proceeding, for completeness, we also look at the case when training samples and validation samples are grouped independently into their respective training-bins and validation-bins. Figure 2(a) compares $C_{train,i}$ in training-bin-$i$ with $C_{val,i}$ in validation-bin-$i$. We observe a similar behaviour as mentioned above. Note that since the binning is independent, the boundaries of training-bin-$i$ may not be exactly the same as that of validation-bin-$i$, however as shown in Figure 11 Appendix C (along with rest of the bins and their bin boundaries), they are quite close, meaning that a training group in lower (/middle/higher) probability region have good correspondence with the validation group in a similar nearby region.

Going forward, for the ease of algorithm design, we will simply stick to the case of "common binning" where training samples are grouped as per validation-bin boundaries. This will allows us to maintain a one-to-one correspondence between the boundaries of the $i$th training and validation group.

## 5    PROPOSED METHOD

Let's denote the $n$th training sample's true class posterior $\hat{p}_{y_{\text{true}}}$ by $p_n$. Given that $p_n$ falls into validation-bin-$b$, our goal is to keep $p_n$, or as per the discussion above its averaged equivalent $C_{train,b}$, closer to $A_{val,b}$ so that the same is reflected on $C_{val,b}$. For manipulating $p_n$, we will utilize the regularization effect that focal loss's parameter $\gamma$ has on the confidence of the predictions Mukhoti

---

[1]It may happen that no training sample belong to a particular validation-bin's boundaries. In that case, $C_{train,i}$ has been shown to drop to zero for example in bin-14 in Figure 2 (b).

et al. (2020). At this point, one can choose to update $\gamma_b$ either based on (1) how far $p_n$ is from $A_{val,b}$ i.e. $\gamma = f(p_n - A_{val,b})$ or (2) how far $C_{val,b}$ is from $A_{val,b}$ i.e. $\gamma = f(C_{val,b} - A_{val,b})$. Such a $\gamma$-update-rule should ensure that whenever the model is over-confident, i.e. $p_n > A_{val,b}$ (or $C_{val,b} > A_{val,b}$), $\gamma$ is increased so that the gradients get smaller which prevents $p_n$ from increasing further. On the other hand, when $p_n < A_{val,b}$ (or $C_{val,b} < A_{val,b}$), i.e. the model is under-confident, we decrease $\gamma$ so as to get larger gradients that in turn will increase $p_n$ [2].

Based on this discussion, next we design and study a calibration-aware $\gamma$-update strategy called CalFocal, which with some additional modifications lead to AdaFocal.

## 5.1 CALIBRATION AWARE FOCAL LOSS (CALFOCAL)

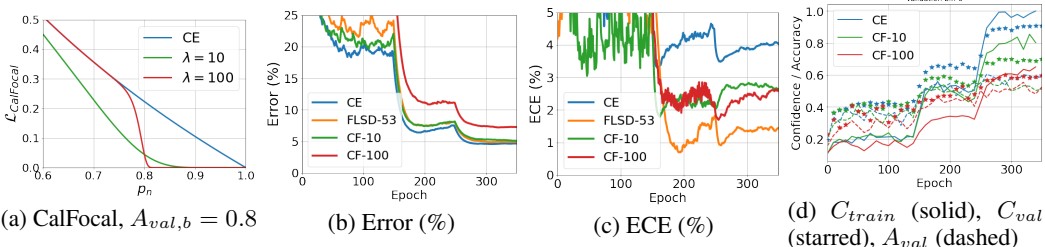

(a) CalFocal, $A_{val,b} = 0.8$      (b) Error (%)      (c) ECE (%)      (d) $C_{train}$ (solid), $C_{val}$ (starred), $A_{val}$ (dashed)

Figure 3: ResNet-50 trained on CIFAR-10 with cross entropy (CE) and CalFocal (CF-$\lambda$). Sub-figure (d) compares $C_{train}$, $C_{val}$ and $A_{val}$ in validation bin-0 to show that as CalFocal tries to keep $C_{train}$ closer to $A_{val}$, $C_{val}$ also gets closer to $A_{val}$.

**Case 1: $\gamma = f(p_n - A_{val,b})$**    Treating $A_{val,b}$ as the point that we want $p_n$ to not deviate from, we make the focal loss parameter $\gamma$ a function of $p_n - A_{val,b}$ to get

$$\mathcal{L}_{CalFocal}(p_n) = -(1-p_n)^{\gamma_n} \log p_n, \qquad \text{with, } \gamma_n = \exp(\lambda(p_n - A_{val,b})), \qquad (1)$$

where, $b$ is the validation-bin in which $p_n$ falls. The hyper-parameter $\lambda$ is the scaling factor which combined with the exponential function helps to quickly ramp up/down $\gamma$. The exponential function adheres to the $\gamma$-*update rule* mentioned earlier and also ensures $\gamma$ is $> 0$. Figure 3(a) plots $\mathcal{L}_{CalFocal}$ vs. $p_n$ for $A_{val,b} = 0.8$. We see that based on the strength of $\lambda$, the loss drastically drops near $p_n = 0.8$ and thereafter remains close to zero. This shows that $\mathcal{L}_{CalFocal}$ aims is to first push $p$ towards 0.8 and then slow its growth towards overconfidence. Next, in Figure 3(c), we find that CalFocal with $\lambda = 10, 100$ is able to reduce the calibration error compared to cross entropy but it is still far from FLSD-53's performance. Also note in Figure 3(b) that too high $\lambda$ (=100) affects the accuracy of model. Most importantly, Figure 3(d) compares $C_{train,i}$ with $C_{val,i}$ (and also $A_{val,i}$) for bin-0, where we find some evidence that the strategy of bringing $p_n$ or $C_{train,i}$ (solid lines) closer to $A_{val,i}$ (dashed lines) results in $C_{val,i}$ (starred lines) getting closer to $A_{val,i}$ as well, thus reducing the calibration error $E_{val,i} = C_{val,i} - A_{val,i}$ slightly.

**Case 2: $\gamma = f(C_{val,b} - A_{val,b})$**    Note that Eq. 1 assigns a different $\gamma_n$ for each training sample. To reduce computation and avoid using a different $\gamma_n$ for each training sample, one can instead use a common $\gamma_b$ for all the training samples that fall into the validation-bin-$b$ by simply making it a function of $C_{val,b} - A_{val,b}$ instead of $p_n - A_{val,b}$.

$$\mathcal{L}_{CalFocal}(p_n) = -(1-p_n)^{\gamma_b} \log p_n, \qquad \text{with, } \gamma_b = \exp(\lambda(C_{val,b} - A_{val,b})) \qquad (2)$$

where, $b$ is the validation-bin in which $p_n$ falls. As shown in Appendix D, it's performance is very similar (or slightly better than) CalFocal in Eq. 1. Further, it makes more sense to update $\gamma$ based on how far $C_{val,b}$ is from $A_{val,b}$ instead of how far $p_n$ is from $A_{val,b}$ because, as shown in Figure 3(d) bin-0, one may find $C_{val,b}$ (starred lines) quite closer to $A_{val,b}$ (dashed lines) even when $p_n$ or its

---

[2]Note that for focal loss increasing $\gamma$ does not always lead to smaller gradients. This mostly holds true in the region $p_n$ approximately $> 0.2$ (see Figure 3(a) in Mukhoti et al. (2020)). However, in practice and as shown by the training-bin boundaries of bin-0 and bin-1 in Figure in Figure 11 Appendix C, we find majority of the training samples to lie above 0.2 during the majority of the training, and therefore, for the experiments in this paper, we simply stick to the rule of increasing $\gamma$ to decrease gradients and stop $p_n$ from increasing and vice versa.

average equivalent $C_{train}$ (solid lines) is far from $A_{val,b}$. At this point when $C_{val,b} = A_{val,b}$, we should stop updating $\gamma$ further, even though $p_n - A_{val,b} \neq 0$, as we have reached our goal of making $E_{val,b} = C_{val,b} - A_{val,b} = 0$. Therefore, we use Eq. 2 of Case 2 as base for AdaFocal.

**Limitations of CalFocal:** (1) Let's say at some point of training, a high $\gamma_b$ over the next few epochs reduces the calibration error $E_{val,b} = C_{val,b} - A_{val,b}$. Then, it is desirable to continue the training with the same high $\gamma_b$. However, note CalFocal's update rule in Eq. 2 which will reduce $\gamma \to 1$ as the $C_{val,b} - A_{val,b} \to 0$. (2) At some point let's say $C_{val,b} - A_{val,b}$ is quite high. This will set $\gamma_b$ to some high value as well depending on the hyper-parameter $\lambda$. Assuming this $\gamma_b$ is still not high enough to bring down the confidence, we would want a way to further increase $\gamma_b$. However, CalFocal is incapable of doing so as it will continue to hold at $\gamma_b = \exp(\lambda(C_{val,b} - A_{val,b}))$. By addressing these two issues in the next sub-section we present the final algorithm for AdaFocal.

## 5.2 Calibration-aware Adaptive Focal loss (AdaFocal)

A straightforward way to address the above limitations is to make $\gamma_{b,t}$ depend on $\gamma_{b,t-1}$ i.e.

$$\mathcal{L}(p_n, t) = -(1 - p_n)^{\gamma_{b,t}} \log p_n, \qquad \text{with, } \gamma_{b,t} = \gamma_{b,t-1} * \exp(C_{val,b} - A_{val,b}). \qquad (3)$$

This update rule address the limitations of CalFocal in the following way. Let's say at some point we observe over-confidence i.e. $E_{val,b} = C_{val,b} - A_{val,b} > 0$. Then, in the next step $\gamma_b$ will be increased. In the subsequent steps, it will continue to increase unless the calibration error $E_{val,b}$ starts decreasing (this additional increase in $\gamma$ was not possible with CalFocal). At this point, if we find $E_{val,b}$ to start decreasing, that would reduce the increase in $\gamma_b$ over the next epochs and $\gamma_b$ will ultimately settle down to a value when $E_{val,b} = 0$ (CalFocal at $E_{val,b} = 0$ will cause $\gamma$ to go down to 1). Next, if this current value of $\gamma_b$ starts causing under-confidence i.e. $C_{val,b} - A_{val,b} < 0$, then the update rule will kick in to reduce $\gamma$ thus allowing $C_{val,b}$ to be increased back to $A_{val,b}$. This oscillating behaviour of AdaFocal around the desired point of $C_{val,b} = A_{val,b}$ is its main adavantage in reducing calibration error in every bin. Additionally, also note the absence of the hyper-parameter $\lambda$ in the exponent of Eq. 3 which makes AdaFocal hyper-parameter free.

Finally, note an undesirable property of Eq. 3 which is the unbounded exponential update. This may easily cause $\gamma_t$ to explode as it can be expanded as $\gamma_t = \gamma_{t-1} \exp(E_{val,t}) = \gamma_0 \exp(E_{val,0} + E_{val,1} + ... + E_{val,t-1} + E_{val,t})$. Thus if $E_{val,t} > 0$ for quite a few number of epochs, $\gamma_t$ will become so large that even if $E_{val,t} < 0$ in the subsequent epochs, it may not decrease to a desired level. We remedy this by simply constraining $\gamma_t$ to an upper bound $\gamma_{\max}$ to get the AdaFocal loss as

$$\mathcal{L}_{AdaFocal}(p_n, t) = -(1 - p_n)^{\gamma_{b,t}} \log p_n, \quad \text{with, } \gamma_{b,t} = \min\{\gamma_{\max}, \gamma_{b,t-1} * e^{C_{val,b} - A_{val,b}}\} \quad (4)$$

An algorithmic description of training with AdaFocal (or CalFocal) is given in Algorithm 1. **Limitation:** One may argue that $\gamma_{\max}$ is again a hyper-parameter; however, note that it does not require any special fine-tuning. Its sole purpose is to stop $\gamma$ from exploding and any reasonable value around 20 works quite well in practice. For all our experiments, we use $\gamma_{\max} = 20$. For comparison of AdaFocal with $\gamma_{\max} = 20$, $\gamma_{\max} = 50$ and unconstrained $\gamma_{\max} = \infty$, please refer to Appendix L.

## 6 Experiments

**Experimental setup** We evaluate the performance of our proposed method on image and text classification tasks. For image classification, we use CIFAR-10, CIFAR-100 Krizhevsky (2009), Tiny-ImageNet Deng et al. (2009), and ImageNet Russakovsky et al. (2015) to analyze the calibration of ResNet50, ResNet-100 He et al. (2016), Wide-ResNet-26-10 Zagoruyko & Komodakis (2016), and DenseNet-121 Huang et al. (2017) models. For text classification, we use the 20 Newsgroup dataset Lang (1995) and train the Global Pooling CNN model Lin et al. (2014). Further details about the datasets, models and experimental configurations are given in Appendix E.

**Baseline** As baseline calibration methods we use MMCE Kumar et al. (2018), Brier loss Brier (1950), Label smoothing Müller et al. (2019) and sample-dependent focal loss FLSD-53. We also report the effect of temperature scaling Guo et al. (2017) on top of these calibration methods. Following Mukhoti et al. (2020), we select the optimal temperature that produces the minimum ECE on the validation set by searching in the interval $(0, 10]$ with step size of $0.1$.

**Results.** In Figure 4, we compare AdaFocal against cross entropy (CE) and FLSD-53 for ResNet-50 trained on various small to large-scale image datasets. We chose FLSD-53 as our competitive baseline

---

**Algorithm 1:** CalFocal, AdaFocal

1   **Input**: $D_{train} = \{(\mathbf{x}_n, y_{true,n})\}_{n=1}^{N_{train}}$ and $D_{val} = \{(\mathbf{x}_n, y_{true,n})\}_{n=1}^{N_{val}}$ ;

2   **Initialization** at $t = 0$: **for** $i = 1$ **to** $M$ **do**

3      $B_{val,t,i} = \left(\frac{i-1}{M}, \frac{i}{M}\right]$          // equally spaced validation-bins;

4      $C_{val,t+1,i} = A_{val,t+1,i} = \frac{2i-1}{2M}$       // mid point of the bin;

5      $\gamma_{t,i} = 1$;

6   **Training**: **for** $t = 0$ **to** $T$ **do**

7      $L_t = 0$;

8      **for** $n = 1$ **to** $N_{train}$ **do**

9          $p_n = f_{w_t}(\mathbf{x}_n)$             // denoting $p_n = p_{y_{true,n}}$;

10        $b = \text{get\_bin\_index}(p_n, \{B_{t,i}\})$      // validation-bin inside which $p$ lies;

11        $L_t \mathrel{+}= -(1 - p_n)^{\gamma_{t,b}} \log p$       // use $\gamma_{t,b}$ of $b$th bin to compute loss;

12      $w_{t+1} = \text{gradient\_update}(w_t, L_t)$;

13      **for** $i = 1$ **to** $M$ **do**        // Using updated model $f_{w_{t+1}}$ on $D_{val}$, update bin statistics and $\gamma$

14        Re-compute bin boundaries $B_{t+1,i}$ and corresponding $C_{val,t+1,i}$, $A_{val,t+1,i}$;

15        **if** *CalFocal* **then**

16          $\gamma_{t+1,i} = \exp(\lambda * (C_{val,t+1,i} - A_{val,t+1,i}))$;

17        **else if** *AdaFocal* **then**

18          $\gamma_{t+1,i} = \min\{\gamma_{\max}, \ \gamma_{t,i} * \exp(C_{val,t+1,i} - A_{val,t+1,i})\}$;

---

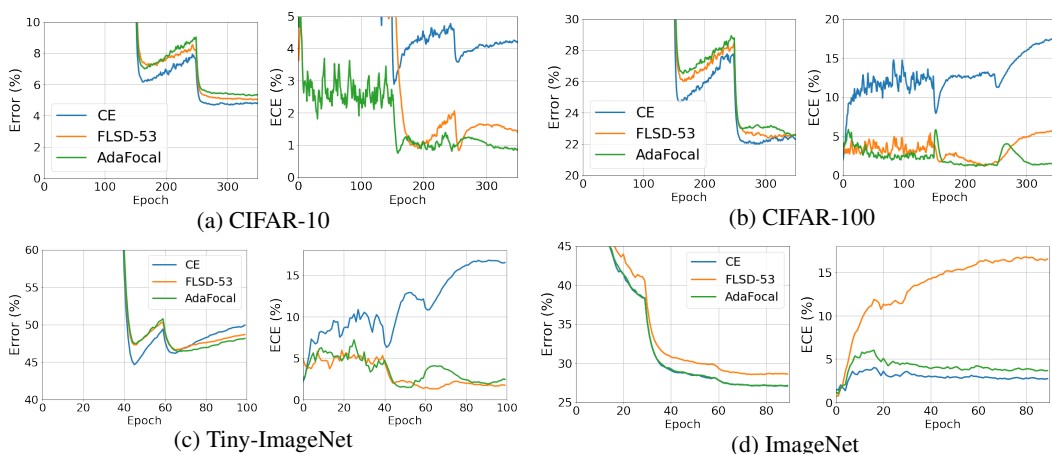

Figure 4: Test set error and calibration of ResNet-50 trained on small to large-scale image datasets with cross entropy (CE), FLSD-53 and AdaFocal. In each subfigure **Left**: Error (%), **Right**: ECE (%). The plots have been averaged over 5 runs. AdaFocal consistently achieves low calibration error across all datasets while maintaining the accuracy.

as it was shown to be consistently better than MMCE, Brier Loss and Label smoothing Mukhoti et al. (2020) across many datasets-model pairs. The figure plots the test set error and ECE calibration metric. In Figure 5, for ResNet-50 on CIFAR-10 and ImageNet, we plot (1) the calibration statistics $E_{val} = C_{val} - A_{val}$ of the validation set and (2) the dynamics of associated $\gamma_t$ used by AdaFocal during the training for a few bins covering lower, middle, and higher probability regions.

From these figures, we first observe that for CIFAR-10, CIFAR-100 and Tiny-ImageNet, FLDS-53 is much better calibrated than CE. This is because, as shown in Figure 5(a) for ResNet-50 and CIFAR-10, CE is over-confident compared to FLSD-53 in every bin. For ImageNet, however, the behaviour is reversed: FLSD-53 is poorly calibrated than CE. The reason, as shown in Figure 5(b), is that due to the use of high values $\gamma = 5, 3$, FLSD-53 makes the model largely under-confident in each bin, leading to an overall high calibration error. This shows that FLSD-53 is a strategy based on heuristic (from a limited number of dataset-model pairs) that does not generalize well. AdaFocal, on the other hand, is well calibrated for all the four dataset-model pairs while achieving similar accuracy.

The dynamics/evolution of $\gamma_t$ during training for different bins is shown in Figure 5: (1) for CIFAR-10, we find $\gamma_t$ to be closer to 1 for higher bins and closer to 20 for lower bin. These $\gamma$s found by AdaFocal result in better calibration than $\gamma = 5, 3$ of FLSD-53. (2) for ImageNet, we find AdaFocal's

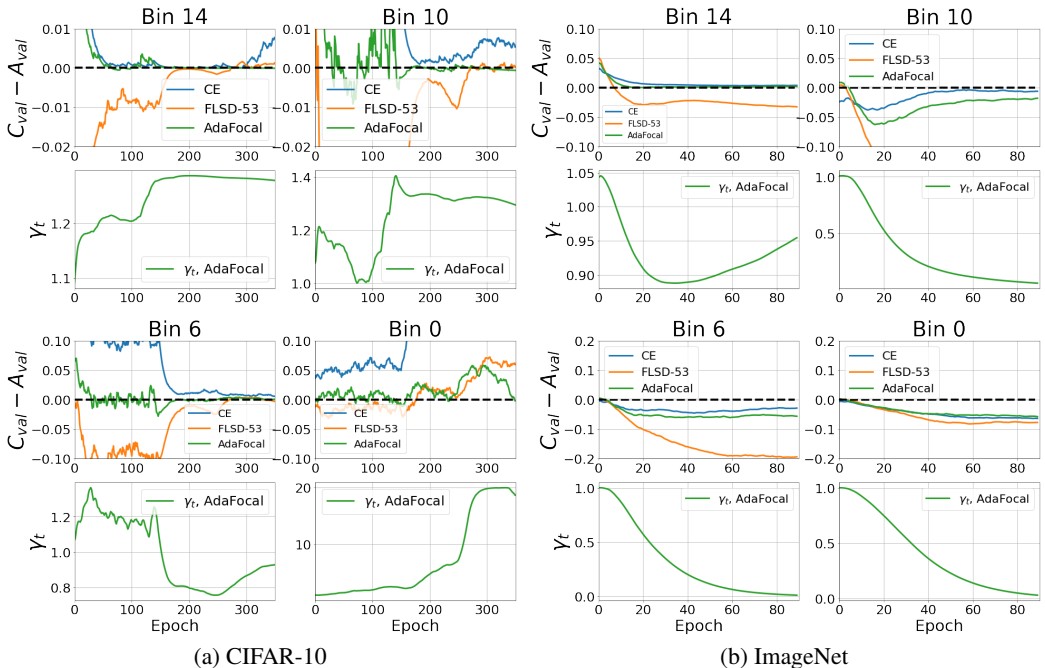

(a) CIFAR-10               (b) ImageNet

Figure 5: Dynamics of $\gamma$ in different bins and calibration statistics of validation set used by AdaFocal for ResNet-50 trained on CIFAR-10 and ImageNet. Each bin has two subplots: **Top**: $E_{val} = C_{val} - A_{val}$, **Bottom**: evolution of $\gamma_t$. Black dotted line in top plot represent zero calibration error. We observe that AdaFocal is able to find the $\gamma$s that produces the lowest calibration in each bin. For CIFAR-10, $\gamma > 1$ whereas for ImageNet $\gamma \to 0$ because, from Figure 4(d), we see that cross entropy ($\gamma = 0$) is better calibrated on ImageNet.

$\gamma \to 0$. This makes sense because for ImageNet, from Figure 4(d), cross entropy (i.e. $\gamma = 0$ for every bin) is much better calibrated than FLSD-53 and AdaFocal (starting from $\gamma = 1$) also ultimately settles down to CE ($\gamma = 0$) to achieve a similar level of calibration. This confirms that during training, unlike CE or FLSD-53, AdaFocal being aware of the network's current under/over-confidence is able to guide the $\gamma$s to the values that maintain a well calibrated model at every step. Also note that for an unseen dataset-model pair there's no way to know beforehand which $\gamma$ will perform better but these empirical evidence show that AdaFocal will automatically find those appropriate $\gamma$s.

Rest of the experiments are shown in Table 1 (ECE) and Table 2 (Error)[3]. From Table 1, we observe that prior to temperature scaling AdaFocal outperforms the baseline methods by a substantial margin in 9 out of 11 cases. With post-temperature scaling included, AdaFocal achieves the lowest calibration error in 7 out of the 11 experiments. Further, observe that in many cases temperature scaling on top of AdaFocal does not offer any improvement (optimal temperature = 1). For the rest, the optimal temperature is close to 1 indicating that AdaFocal produces innately calibrated models during training itself. The consistency of AdaFocal across other calibration metrics is shown through AdaECE and classwise-ECE in Appendix F. $ECE_{debias}$ (15 and 30 bins), $ECE_{EW-sweep}$ (equal-width), and $ECE_{EM-sweep}$ (equal-mass) are reported in Appendix G. Significance of the results is confirmed through ECE error bars with mean and standard deviations computed over 5 runs in Appendix H.

**Number of bins** The ECE metrics in the paper are reported using 15 bins. For AdaFocal training we experiment with 5, 10, 15, 20, 30, and 50 equal-mass (adaptive) binning when drawing calibration statistics form the validation set as reported in Appendix I. We find the best results to be from the range 10 to 20. Performance degrades when the number of bins are too small ($< 10$) or too large ($> 20$), therefore, for the AdaFocal training in the paper we use 15 bins as well.

**Out-of-Distribution (OOD) detection.** Following Mukhoti et al. (2020), we report the performance of AdaFocal on an OOD detection task. We train ResNet-110 and Wide-ResNet26-10 on

---

[3]While reproducing the baseline experiments in Mukhoti et al. (2020) we obtained very similar results, therefore, we simply borrow the exact values to maintain consistent comparison.

| Dataset | Model | Cross Entropy | | Brier Loss | | MMCE | | LS-0.05 | | FLSD-53 | | AdaFocal | |
|---|---|---|---|---|---|---|---|---|---|---|---|---|---|
| | | Pre T | Post T | Pre T | Post T | Pre T | Post T | Pre T | Post T | Pre T | Post T | Pre T | Post T |
| CIFAR-10 | ResNet-50 | 4.35 | 1.35(2.5) | 1.82 | 1.08(1.1) | 4.56 | 1.19(2.6) | 2.96 | 1.67(0.9) | 1.55 | 0.95(1.1) | 0.8 | **0.65**(1.08) |
| | ResNet-110 | 4.41 | 1.09(2.8) | 2.56 | 1.25(1.2) | 5.08 | 1.42(2.8) | 2.09 | 2.09(1.0) | 1.87 | 1.07(1.1) | 0.8 | **0.65**(1.06) |
| | Wide-ResNet-26-10 | 3.23 | 0.92(2.2) | 1.25 | 1.25(1.0) | 3.29 | 0.86(2.2) | 4.26 | 1.84(0.8) | 1.56 | 0.84(0.9) | **0.7** | 0.7(1.0) |
| | DenseNet-121 | 4.52 | 1.31(2.4) | 1.53 | 1.53(1.0) | 5.1 | 1.61(2.5) | 1.88 | 1.82(0.9) | 1.22 | 1.22(1.0) | 0.76 | **0.66**(1.02) |
| CIFAR-100 | ResNet-50 | 17.52 | 3.42(2.1) | 6.52 | 3.64(1.1) | 15.32 | 2.38(1.8) | 7.81 | 4.01(1.1) | 4.5 | 2.0(1.1) | **1.3** | 1.3(1.0) |
| | ResNet-110 | 19.05 | 4.43(2.3) | 7.88 | 4.65(1.2) | 19.14 | 3.86(2.3) | 11.02 | 5.89(1.1) | 8.56 | 4.12(1.2) | **1.3** | 1.3(1.0) |
| | Wide-ResNet-26-10 | 15.33 | 2.88(2.2) | 4.31 | 2.7(1.1) | 13.17 | 4.37(1.9) | 4.84 | 4.84(1) | 3.03 | **1.64**(1.1) | 1.92 | 1.92(1.0) |
| | DenseNet-121 | 20.98 | 4.27(2.3) | 5.17 | 2.29(1.1) | 19.13 | 3.06(2.1) | 12.89 | 7.52(1.2) | 3.73 | **1.31**(1.1) | 1.74 | 1.74(1.0) |
| Tiny-ImageNet | Resnet-50 | 15.32 | 5.48(1.4) | 4.44 | 4.13(0.9) | 13.01 | 5.55(1.3) | 15.23 | 6.51(0.7) | 1.76 | **1.76**(1) | 2.41 | 2.25(0.96) |
| ImageNet | ResNet-50 | 2.81 | 1.58(0.9) | - | - | - | - | - | - | 16.77 | 2.52(0.7) | 3.68 | **1.10**(0.9) |
| 20 Newsgroups | Global-pool CNN | 17.92 | 2.39(3.4) | 13.58 | 3.22(2.3) | 15.48 | 6.78(2.2) | 4.79 | 2.54(1.1) | 6.92 | **2.19**(1.5) | 2.72 | 2.67(1.12) |

Table 1: Test set ECE(%) for different methods (pre and post temperature scaling). Underlined values mark the lowest error among Pre-T results and bold marks the overall lowest in the row. Optimal temperatures are given in brackets. For AdaFocal, the values are averaged over 5 runs.

| Dataset | Model | Cross Entropy | Brier Loss | MMCE | LS-0.05 | FLSD-53 | AdaFocal |
|---|---|---|---|---|---|---|---|
| CIFAR-10 | ResNet-50 | **4.95** | 5.0 | 4.99 | 5.29 | 4.98 | 5.30 |
| | ResNet-110 | **4.89** | 5.48 | 5.4 | 5.52 | 5.42 | 5.27 |
| | Wide-ResNet-26-10 | **3.86** | 4.08 | 3.91 | 4.2 | 4.01 | 4.5 |
| | DenseNet-121 | **5.0** | 5.11 | 5.41 | 5.09 | 5.46 | 5.2 |
| CIFAR-100 | ResNet-50 | 23.3 | 23.39 | 23.2 | 23.43 | 23.22 | **22.60** |
| | ResNet-110 | 22.73 | 25.1 | 23.07 | 23.43 | **22.51** | 22.79 |
| | Wide-ResNet-26-10 | 20.7 | 20.59 | 20.73 | 21.19 | 20.11 | **20.07** |
| | DenseNet-121 | 24.52 | 23.75 | 24.0 | 24.05 | 22.67 | **22.22** |
| Tiny-ImageNet | Resnet-50 | 49.81 | 53.2 | 51.31 | **47.12** | 49.06 | 48.26 |
| ImageNet | Resnet-50 | **27.08** | - | - | - | 28.53 | 27.22 |
| 20 Newsgroups | Global-pool CNN | 26.68 | 27.06 | 27.23 | **26.03** | 27.98 | 28.53 |

Table 2: Test set error (%). The model with the lowest error is marked in bold.

CIFAR-10 as the in-distribution data and test on SVHN Netzer et al. (2011) and CIFAR-10-C Hendrycks & Dietterich (2019) (with level 5 Gaussian noise corruption) as OOD data. Using entropy of the softmax as the measure of uncertainty, the corresponding ROC plots are shown in Figure 6 and AUROC scores are reported in Table 10 in Appendix J. We see that models trained with AdaFocal outperform focal loss $\gamma = 3$ (FL-3) and FLSD-53. For the exact AUROC scores, please refer to Appendix J. These results further highlight the benefit of an inherently calibrated model produced using AdaFocal as post-hoc techniques such as temperature scaling, as shown in the figure, is ineffective under distributional shift Snoek et al. (2019).

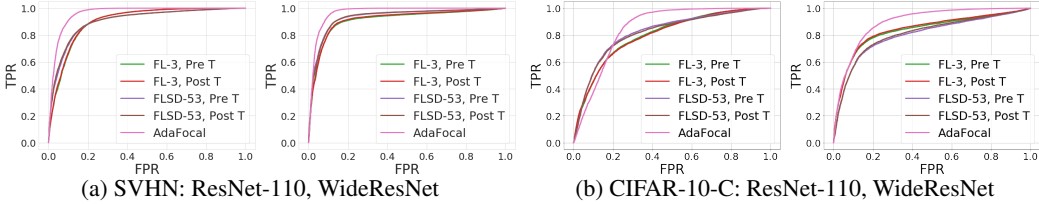

(a) SVHN: ResNet-110, WideResNet   (b) CIFAR-10-C: ResNet-110, WideResNet

Figure 6: ROC for ResNet-110 and Wide-ResNet-26-10 trained on in-distribution CIFAR-10 and tested on out-of-distribution (a) SVHN and (b) CIFAR-10-C. FL-3 refers to Focal loss $\gamma = 3$, Pre/Post T refers to before and after temperature scaling respectively.

## 7 CONCLUSION

In this work, we first revisit the calibration properties of regular focal loss and highlight the downside of using a fixed $\gamma$ for all samples. Particularly, by studying the calibration behaviour of different samples in different probability region, we find that there's no single $\gamma$ that achieves the best calibration over the entire region. We use this observation to motivate the selection of $\gamma$ independently for each sample (or group of samples) based on the knowledge of network's under/over-confidence. We propose a calibration-aware adaptive focal loss called AdaFocal that accounts for such information and updates the $\gamma_t$ at every step based on $\gamma_{t-1}$ from the previous step and the magnitude of network's under/over-confidence. We find AdaFocal to perform consistently better across different dataset-model pairs producing innately calibrated models that most times do not substantially benefit from post-hoc processing of temperature scaling. Additionally, we find models trained with AdaFocal to be significantly better in out-of-distribution detection task.

**Reproducibility**    For reproducibility, we have include in the supplementary material a zip file that contains the code base for running the experiments. For running particular experiments

- CIFAR-10, ResNet-50, Cross entropy: *python train.py –dataset cifar10 –model resnet50 –loss cross_entropy –num_bins 15 -e 400 –save-path experiments/cifar10_resnet50_ce*

- CIFAR-100, ResNet-50, Cross entropy: *python train.py –dataset cifar100 –model resnet50 –loss cross_entropy –num_bins 15 -e 400 –save-path experiments/cifar100_resnet50_ce*

- Tiny-ImageNet, ResNet-50, Cross entropy: *python train.py –dataset tiny_imagenet –model resnet50_ti –loss cross_entropy –num_bins 15 –first-milestone 40 –second-milestone 60 -e 100 -b 64 -tb 64 –dataset-root data/tiny-imagenet-200 –save-path experiments/tinyImageNet_resnet50_ce*

- 20 Newgroups, CNN, Cross entropy: *python main.py –loss cross_entropy –num-epochs 50 –num-bins 15 –save-path experiments/cnn_ce*

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

## APPENDICES

## A ADAFOCAL'S GENERALIZATION TO LARGE SCALE DATASET (IMAGENET)

For ImageNet, FLSD-53 seems to perform very poorly in terms of calibration. The reason is that due to higher values of $\gamma = 5, 3$ FLSD-53 becomes extremely under-confident in each bin leading to a high calibration error. AdaFocal, on the hand, remains well calibrated which confirms that during training, unlike CE or FLSD-53, AdaFocal being aware of the network's current under/over-confidence (through the validation set) is able to adjusts the $\gamma$s in a way that maintains a well calibrated model at every step. Further, in Figure 8, note the dynamics/evolution of $\gamma_t$ in different bins. For ImageNet, we find AdaFocal's $\gamma \to 0$ which makes sense because, from Figure 7, cross entropy (i.e. $\gamma = 0$ for every bin) is much better calibrated than FLSD-53 and AdaFocal (starting from $\gamma = 1$) settles down as CE ($\gamma = 0$). Note that for an unseen dataset-model pair it is not possible to know beforehand whether CE or focal loss will perform better. However, from these experiments, we find strong evidence that, for any dataset-model pair, AdaFocal will lead to the $\gamma$s that result in the best calibration.

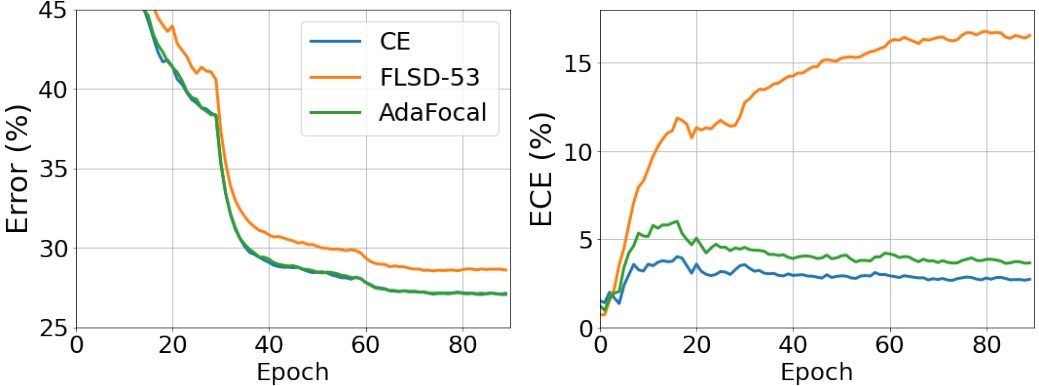

Figure 7: Test set error and calibration of ResNet-50 trained on ImageNet. **Left**: Error, **Right**: ECE.

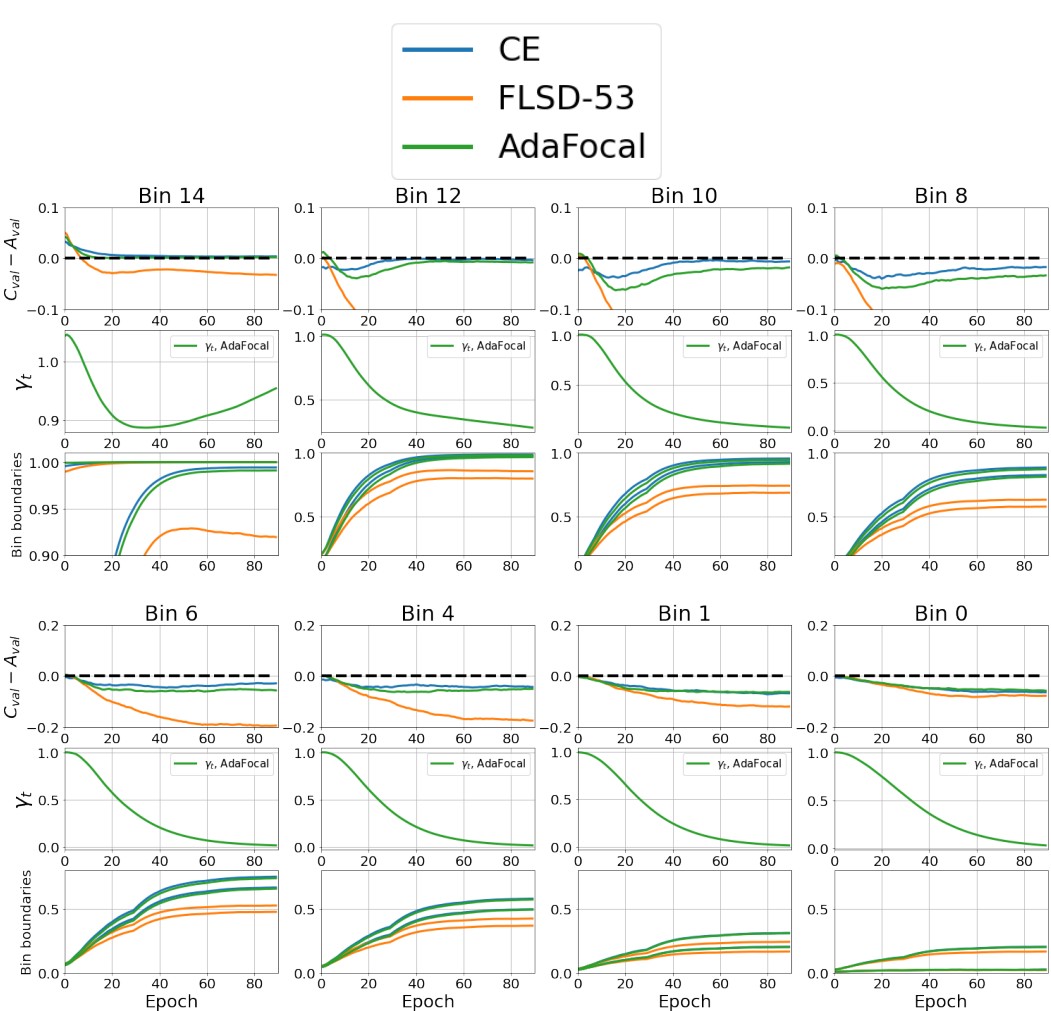

Figure 8: Dynamics of $\gamma$ and calibration in different bins when ResNet-50 is trained on ImageNet. Each bin has three subplots: **top**: $E_{val,i} = C_{val,i} - A_{val,i}$, **middle**: evolution of $\gamma_t$, and **bottom**: bin boundaries. Black dotted line in top plot represent zero calibration error. We observe that for most of the bins AdaFocal approaches $\gamma \to 0$ (which is the CE entropy loss) that result in the best calibration.

# B    CALIBRATION BEHAVIOUR OF FOCAL LOSS IN DIFFERENT BINS

In the main paper, we showed the calibration behavior of different focal losses for ResNet50 trained on CIFAR-10 for only a few bins. For completeness, the rest of the bins and their calibration error $E_i = C_{val,i} - A_{val,i}$ are shown in Figure 9 for focal losses with $\gamma = 0, 3, 4, 5$. We observe that there's no single $\gamma$ that performs the best across all the bins. Rather, every bin seems to have a particular $\gamma$ that achieves the best calibration.

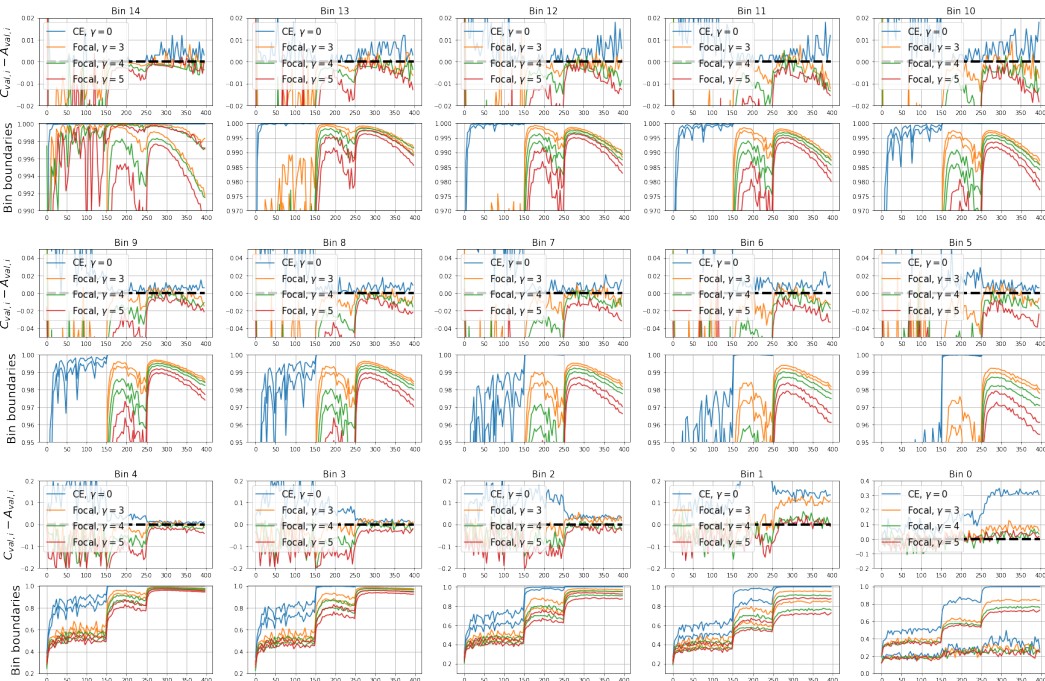

Figure 9: ResNet50 model trained on CIFAR-10 using different focal losses $\gamma = 0, 3, 4, 5$. **Top**: $E_{val,i} = C_{val,i} - A_{val,i}$, **Bottom**: bin boundaries. The statistics are computed on the validations set $(5,000$ examples) using 15 equal-mass bins. The black horizontal line in top subfigure represents zero calibration error $E_{val,i} = 0$.

## C CORRESPONDENCE BETWEEN CONFIDENCE OF TRAINING AND VALIDATION SAMPLES

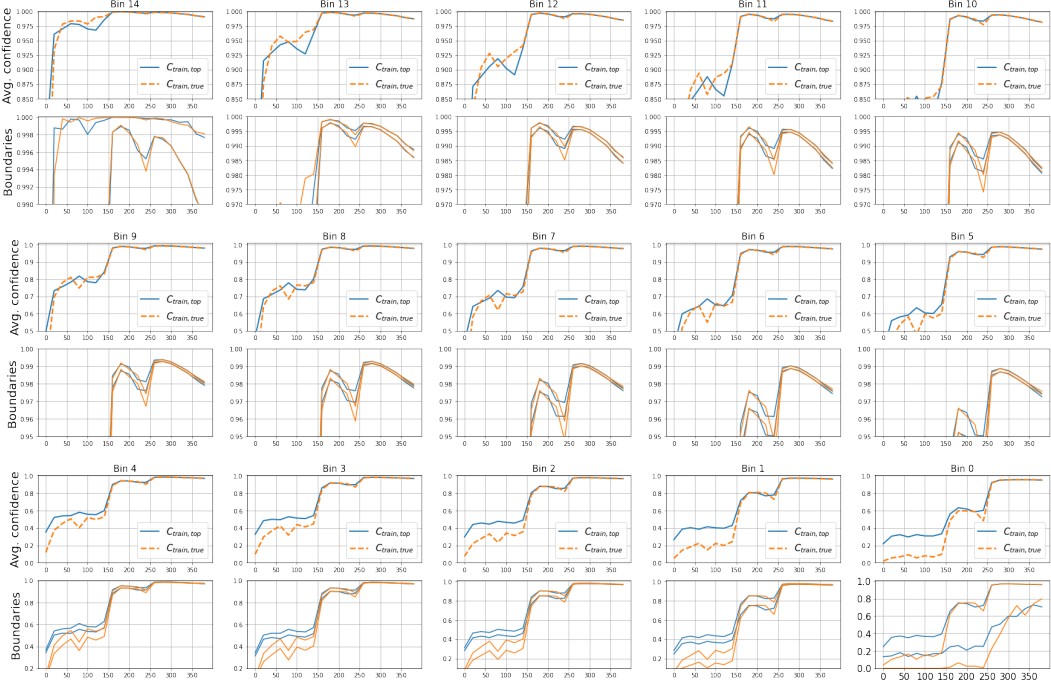

Figure 10: ResNet50 trained on CIFAR-10 with focal loss $\gamma = 3$. It shows that $C_{train,true,i}$ and $C_{train,top,i}$ are almost the same during major part of the training. This is because as the model approaches towards 100% accuracy on the training set, the top predicted class and the true class for a training sample become the same.

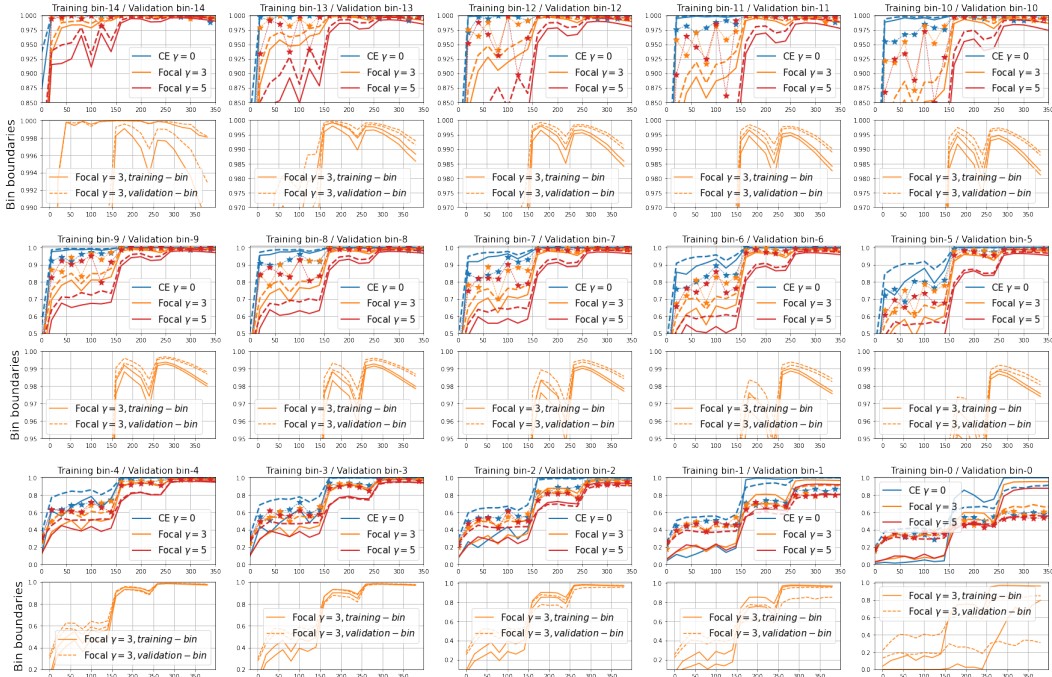

Figure 11: **Independent binning**: training samples and validation samples are grouped independently into training-bin and validation-bin respectively. The top subfigure for each bin shows the correspondence between average confidence of a group of training samples $C_{train,true,i}$ and a group of validation samples $C_{val,top,i}$ when ResNet-50 is trained on CIFAR-10 with focal loss $\gamma = 0, 3, 5$. The binning is adaptive with 15 equal-mass bins. **Solid line**: $C_{train,true,i}$ in training-bin $i$, **Dashed line**: $C_{val,top,i}$ and **Star-dashed line**: $A_{val,i}$ in validation-bin $i$. The bottom subfgure shows the bin binudary only for focal loss $\gamma = 3$.

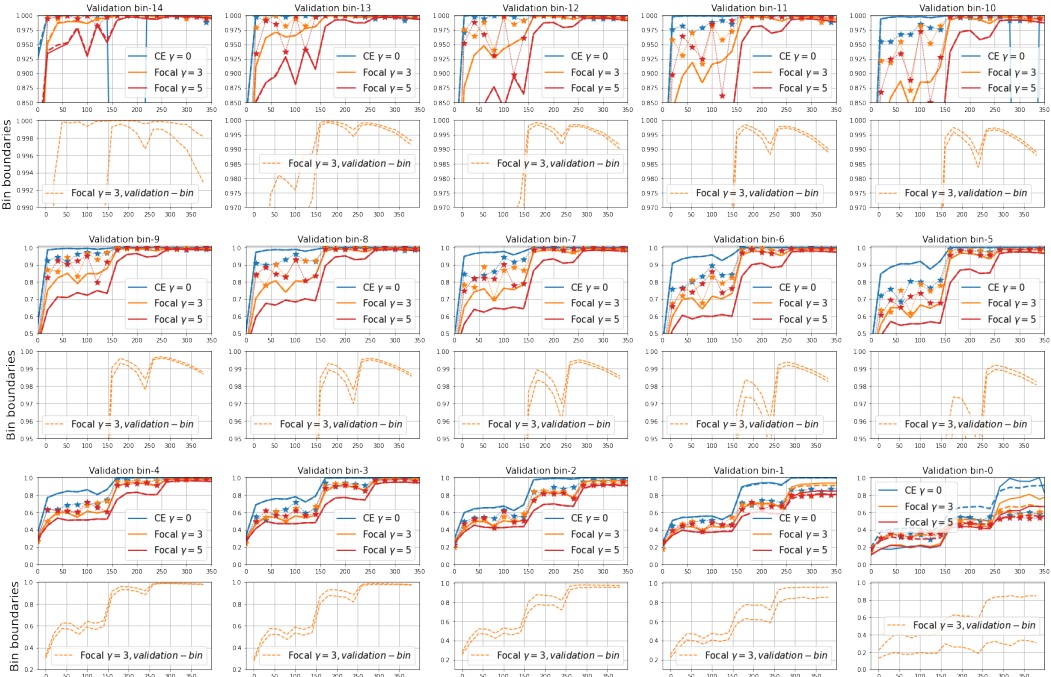

Figure 12: **Common binning**: training samples are grouped using the bin boundaries of the validation-bins. The top subfigure for each bin shows the correspondence between average confidence of a group of training samples $C_{train,true,i}$ and a group of validation samples $C_{val,top,i}$ when ResNet-50 is trained on CIFAR-10 with focal loss $\gamma = 0, 3, 5$. The binning is adaptive with 15 equal-mass bins. **Solid line**: $C_{train,true,i}$ in validation-bin $i$, **Dashed line**: $C_{val,top,i}$ and **Star-dashed line**: $A_{val,i}$ in validation-bin $i$. The bottom subfgure shows the bin binudary only for focal loss $\gamma = 3$.

# D    CALFOCAL LOSS

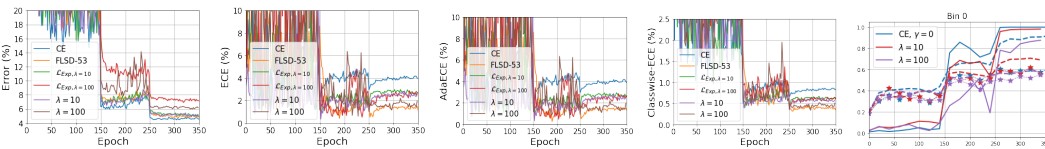

Figure 13: ResNet-50 trained on CIFAR-10 using CalFocal loss. $\mathcal{L}_{Exp,\lambda}$ = CalFocal loss in Eq. 1 of the main paper. Legend $\lambda$ = CalFocal loss in Eq. 2 with common $\gamma_b$ for all training samples in bin $b$. Ce refers to cross entropy.

# E    DATASETS AND EXPERIMENTS

## E.1    DATASET DESCRIPTION

**CIFAR-10** Krizhevsky (2009): This dataset contains $60,000$ coloured images of size $32 \times 32$, which are equally divided into 10 classes. A split of $45,000/5,000/10,000$ images is used as train/validation/test sets respectively.

**CIFAR-100** Krizhevsky (2009): This dataset contains $60,000$ coloured images of size $32 \times 32$, which are equally divided into 100 classes. A split of $45,000/5,000/10,000$ images is used as train/validation/test sets respectively.

**ImageNet** Russakovsky et al. (2015): ImageNet Large Scale Visual Recognition Challenge (ILSVRC) 2012-2017 is an image classification and localization dataset. This dataset spans 1000 object classes and contains 1,281,167 training images and 50,000 validation images.

**Tiny-ImageNet** Deng et al. (2009): It is a subset of the ImageNet dataset with $64 \times 64$ dimensional images and 200 classes. It has 500 images per class in the training set and 50 images per class in the validation set.

**20 Newsgroups** Lang (1995): This dataset contains $20,000$ news articles, categorised evenly into 20 different newsgroups. Some of the newsgroups are very closely related to each other (e.g. comp.sys.ibm.pc.hardware / comp.sys.mac.hardware), while others are highly unrelated (e.g misc.forsale / soc.religion.christian). We use a train/validation/test split of $15,098/900/3,999$ documents.

## E.2    EXPERIMENTAL DETAILS

For all our experiments, we have used Nvidia Titan X Pascal GPU with 12GB memory.

**CIFAR-10** and **CIFAR-100**: We use SGD with a momentum of 0.9 as our optimiser, and train the networks for 350 epochs, with a learning rate of 0.1 for the first 150 epochs, 0.01 for the next 100 epochs, and 0.001 for the last 100 epochs. We use a training batch size of 128. The training data is augmented by applying random crops and random horizontal flips.

**Tiny-ImageNet**: We use SGD with a momentum of 0.9 as our optimiser, and train the models for 100 epochs with a learning rate of 0.1 for the first 40 epochs, 0.01 for the next 20 epochs and 0.001 for the last 40 epochs. We use a training batch size of 64. Note that we use 50 samples per class (i.e. a total of 10000 samples) from the training set as the validation set. Hence, the training is only on 90000 images. We use the Tiny-ImageNet validation set as our test set.

**ImageNet**: We use SGD as our optimiser with momentum of 0.9 and weight decay $10^{-4}$, and train the models for 90 epochs with a learning rate of 0.01 for the first 30 epochs, 0.001 for the next 30 epochs and 0.0001 for the last 30 epochs. We use a training batch size of 128. We divide the 50,000 validation images into validation and test set of 25,000 images each.

**20 Newsgroups**: We train the Global Pooling CNN Network Lin et al. (2014) using the Adam optimiser, with learning rate 0.001, and default betas 0.9 and 0.999. We used Glove word embeddings

Pennington et al. (2014) to train the network. We train the model for 50 epochs and use the model at the end to evaluate the performance.

All our experiments are implemented in PyTorch. The hyperparameters that are not explicitly mentioned above are set to their default values. For CIFAR-10/100 and Tiny-ImageNet, AdaFocal is implemented on top of the base code available from Mukhoti (2020). The code for 20 Newsgroups is implemented in PyTorch by adapting the code (TensorFlow) available from Kumar (2018).

The experimental results in the paper are reported for the model at the end of (1) CIFAR-10/100: 350 epochs, (2) Tiny-ImageNet: 100 epochs, (3) ImageNet: 90 epochs, and (4) 20 NewsGroups: 50 epochs.

## F  ADAECE AND CLASSWISE-ECE PERFORMANCE

Here, we compare the performance of AdaFocal against the baseline methods in terms of AdaECE and classwise-ECE in Table 3 and 4 respectively. For CIFAR-10/100, the values are reported for the model at the end of 350 epochs; for Tiny-ImageNet, at the end of 100 epochs; and for 20 NewsGroup dataset, at the end of 50 epochs. From these tables, we observe that AdaFocal outperforms all the baseline methods by a substantial margin, especially if we compare the pre-temperature scaling results.

| Dataset | Model | Cross Entropy | | Brier Loss | | MMCE | | LS-0.05 | | FLSD-53 | | AdaFocal | |
|---|---|---|---|---|---|---|---|---|---|---|---|---|---|
| | | Pre T | Post T | Pre T | Post T | Pre T | Post T | Pre T | Post T | Pre T | Post T | Pre T | Post T |
| CIFAR-10 | ResNet-50 | 4.33 | 2.14(2.5) | 1.74 | 1.23(1.1) | 4.55 | 2.16(2.6) | 3.89 | 2.92(0.9) | 1.56 | 1.26(1.1) | __0.56__ | 0.88(0.9) |
| | ResNet-110 | 4.4 | 1.99(2.8) | 2.6 | 1.7(1.2) | 5.06 | 2.52(2.8) | 4.44 | 4.44(1) | 2.07 | 1.67(1.1) | __0.44__ | 0.44(1.0) |
| | Wide-ResNet-26-10 | 3.23 | 1.69(2.2) | 1.7 | 1.7(1) | 3.29 | 1.6(2.2) | 4.27 | 2.44(0.8) | 1.52 | 1.38(0.9) | __0.64__ | **0.42**(1.1) |
| | DenseNet-121 | 4.51 | 2.13(2.4) | 2.03 | 2.03(1) | 5.1 | 2.29(2.5) | 4.42 | 3.33(0.9) | 1.42 | 1.42(1) | __0.54__ | 0.54(1.0) |
| CIFAR-100 | ResNet-50 | 17.52 | 3.42(2.1) | 6.52 | 3.64(1.1) | 15.32 | 2.38(1.8) | 7.81 | 4.01(1.1) | 4.5 | 2.0(1.1) | __**1.72**__ | 1.72(1.0) |
| | ResNet-110 | 19.05 | 5.86(2.3) | 7.73 | 4.53(1.2) | 19.14 | 4.85(2.3) | 11.12 | 8.59(1.1) | 8.55 | 3.96(1.2) | __1.57__ | 1.57(1.0) |
| | Wide-ResNet-26-10 | 15.33 | 2.89(2.2) | 4.22 | 2.81(1.1) | 13.16 | 4.25(1.9) | 5.1 | 5.1(1) | 2.75 | **1.63**(1.1) | __2.22__ | 2.22(1.0) |
| | DenseNet-121 | 20.98 | 5.09(2.3) | 5.04 | 2.56(1.1) | 19.13 | 3.07(2.1) | 12.83 | 8.92(1.2) | 3.55 | **1.24**(1.1) | __1.54__ | 1.54(1.0) |
| Tiny-ImageNet | ResNet-50 | 15.23 | 5.41(1.4) | 4.37 | 4.07(0.9) | 13.0 | 5.56(1.3) | 15.28 | 6.29(0.7) | 1.42 | 1.42(1) | __1.26__ | 1.26(1.0) |
| ImageNet | ResNet-50 | __2.93__ | 1.5(0.9) | - | - | - | - | - | - | 16.77 | 2.62(0.7) | 3.73 | **0.84**(0.9) |
| 20 Newsgroups | Global-pool CNN | 17.91 | **2.23(3.4)** | 13.57 | 3.11(2.3) | 15.21 | 6.47(2.2) | 4.39 | 2.63(1.1) | 6.92 | 2.35(1.5) | __2.38__ | 2.38(1.0) |

Table 3: AdaECE(%). Underline marks the lowest error among pre-temperature scaling values. Bold marks the overall lowest in the row.

| Dataset | Model | Cross Entropy | | Brier Loss | | MMCE | | LS-0.05 | | FLSD-53 | | AdaFocal | |
|---|---|---|---|---|---|---|---|---|---|---|---|---|---|
| | | Pre T | Post T | Pre T | Post T | Pre T | Post T | Pre T | Post T | Pre T | Post T | Pre T | Post T |
| CIFAR-10 | ResNet-50 | 0.91 | 0.45(2.5) | 0.46 | 0.42(1.1) | 0.94 | 0.52(2.6) | 0.71 | 0.51(0.9) | 0.42 | 0.42(1.1) | __0.29__ | 0.32(0.9) |
| | ResNet-110 | 0.91 | 0.50(2.8) | 0.59 | 0.50(1.2) | 1.04 | 0.55(2.8) | 0.66 | 0.66(1) | 0.48 | 0.44(1.1) | __0.33__ | 0.33(1.0) |
| | Wide-ResNet-26-10 | 0.68 | 0.37(2.2) | 0.44 | 0.44(1) | 0.70 | 0.35(2.2) | 0.80 | 0.45(0.8) | 0.41 | 0.31(0.9) | __0.27__ | 0.28(1.1) |
| | DenseNet-121 | 0.92 | 0.47(2.4) | 0.46 | 0.46(1) | 1.04 | 0.57(2.5) | 0.60 | 0.50(0.9) | 0.41 | 0.41(1) | __0.30__ | 0.30(1.0) |
| CIFAR-100 | ResNet-50 | 0.38 | 0.22(2.1) | 0.22 | **0.20(1.1)** | 0.34 | 0.21(1.8) | 0.23 | 0.21(1.1) | __0.20__ | 0.20(1.1) | 0.20 | 0.20(1.0) |
| | ResNet-110 | 0.41 | 0.21(2.3) | 0.24 | 0.23(1.2) | 0.42 | 0.22(2.3) | 0.26 | 0.22(1.1) | 0.24 | 0.21(1.2) | __0.19__ | 0.19(1.0) |
| | Wide-ResNet-26-10 | 0.34 | 0.20(2.2) | 0.19 | 0.19(1.1) | 0.31 | 0.20(1.9) | 0.21 | 0.21(1) | __0.18__ | 0.19(1.1) | 0.19 | 0.19(1.0) |
| | DenseNet-121 | 0.45 | 0.23(2.3) | 0.20 | 0.21(1.1) | 0.42 | 0.24(2.1) | 0.29 | 0.24(1.2) | __0.19__ | 0.20(1.1) | 0.20 | 0.20(1.0) |
| Tiny-ImageNet | ResNet-50 | 0.22 | **0.16(1.4)** | __0.16__ | 0.16(0.9) | 0.21 | **0.16(1.3)** | 0.21 | 0.17(0.7) | __0.16__ | 0.16(1) | __0.16__ | 0.16(1.0) |
| ImageNet | ResNet-50 | __0.03__ | 0.03(0.9) | - | - | - | - | - | - | 0.05 | 0.04(0.7) | __0.03__ | 0.03(0.9) |
| 20 Newsgroups | Global-pool CNN | 1.95 | 0.83(3.4) | 1.56 | **0.82(2.3)** | 1.77 | 1.10(2.2) | __0.93__ | 0.91(1.1) | 1.40 | 1.19(1.5) | 1.02 | 1.02(1.0) |

Table 4: ClasswiseECE(%). Underline marks the lowest error among pre-temperature scaling values. Bold marks the overall lowest in the row.

## G    DEBISED ESTIMATES OF ECE

| Dataset | Model | Cross Entropy | | FLSD-53 | | AdaFocal | |
|---|---|---|---|---|---|---|---|
| | | Pre T | Post T | Pre T | Post T | Pre T | Post T |
| CIFAR-10 | ResNet-50 | 4.05 | 1.7(2.5) | 1.62 | 1.62(1) | 0.47 | 0.82(0.9) |
| | ResNet-110 | 4.38 | 2.2(2.7) | 1.82 | 1.3(1.1) | 0.32 | 0.32(1) |
| | Wide-ResNet-26-10 | 3.52 | 1.89(2.2) | 2.01 | 1.5(0.9) | 0.59 | 0.25(1.1) |
| | DenseNet-121 | 4.26 | 2.15(2.3) | 1.56 | 1.93(0.9) | 0.42 | 0.42(1) |
| CIFAR-100 | ResNet-50 | 17.73 | 3.86(2.2) | 5.52 | 2.92(1.1) | 1.46 | 1.46(1) |
| | ResNet-110 | 19.44 | 6.01(2.3) | 7.31 | 3.55(1.2) | 1.35 | 1.35(1) |
| | Wide-ResNet-26-10 | 14.91 | 3.32(2.1) | 2.53 | 2.53(1) | 2.12 | 2.12(1) |
| | DenseNet-121 | 19.82 | 3.44(2.3) | 2.29 | 2.12(1.1) | 1.27 | 1.27(1) |
| Tiny-ImageNet | ResNet-50 | 16.16 | 5.44(1.5) | 2.00 | 2.00(1) | 0.84 | 0.84 (1) |
| ImageNet | ResNet-50 | 2.89 | 1.42(0.9) | 16.76 | 2.58(0.7) | 3.7 | 0.61(0.9) |
| 20 Newsgroups | Global-pool CNN | 18.36 | 5.23(4.1) | 8.94 | 0.94(1.6) | 1.84 | 1.84(1) |

Table 5: $ECE_{debias}(\%)$ 15 bins. Underline marks the lowest error among pre-temperature scaling values. Bold marks the overall lowest in the row. Optimal temperature is selected based on the lowest ECE on the validation set.

| Dataset | Model | Cross Entropy | | FLSD-53 | | AdaFocal | |
|---|---|---|---|---|---|---|---|
| | | Pre T | Post T | Pre T | Post T | Pre T | Post T |
| CIFAR-10 | ResNet-50 | 4.03 | 1.73(2.5) | 1.57 | 1.57(1) | 0.46 | 0.93(0.9) |
| | ResNet-110 | 4.38 | 2.21(2.7) | 1.81 | 1.23(1.1) | 0.26 | 0.26(1) |
| | Wide-ResNet-26-10 | 3.5 | 1.84(2.2) | 1.98 | 1.5(0.9) | 0.66 | 0.24(1.1) |
| | DenseNet-121 | 4.24 | 2.16(2.3) | 1.6 | 1.95(0.9) | 0.64 | 0.64(1) |
| CIFAR-100 | ResNet-50 | 17.71 | 3.66(2.2) | 5.52 | 2.99(1.1) | 1.55 | 1.55(1) |
| | ResNet-110 | 19.43 | 6.31(2.3) | 7.23 | 3.86(1.2) | 1.17 | 1.17(1) |
| | Wide-ResNet-26-10 | 14.9 | 3.4(2.1) | 2.46 | 2.46(1) | 2.12 | 2.12(1) |
| | DenseNet-121 | 19.81 | 3.49(2.3) | 2.18 | 2.31(1.1) | 1.08 | 1.08(1) |
| Tiny-ImageNet | ResNet-50 | 16.11 | 5.5(1.5) | 1.81 | 1.81(1) | 0.64 | 0.64(1) |
| ImageNet | ResNet-50 | 2.84 | 1.39(0.9) | 16.76 | 2.55(0.7) | 3.67 | 0.91(0.9) |
| 20 Newsgroups | Global-pool CNN | 18.39 | 5.51(4.1) | 8.98 | 1.89(1.6) | 1.92 | 1.92(1) |

Table 6: $ECE_{debias}(\%)$ 30bins. Underline marks the lowest error among pre-temperature scaling values. Bold marks the overall lowest in the row. Optimal temperature is selected based on the lowest ECE on the validation set.

| Dataset | Model | Cross Entropy | | FLSD-53 | | AdaFocal | |
|---|---|---|---|---|---|---|---|
| | | Pre T | Post T | Pre T | Post T | Pre T | Post T |
| CIFAR-10 | ResNet-50 | 4.05 | 1.17(2.5) | 1.31 | 1.31(1) | 0.48 | 0.92(0.9) |
| | ResNet-110 | 4.38 | 1.34(2.7) | 1.96 | 1.05(1.1) | 0.5 | 0.5(1) |
| | Wide-ResNet-26-10 | 3.53 | 1.14(2.2) | 1.22 | 0.92(0.9) | 0.64 | 0.36(1.1) |
| | DenseNet-121 | 4.26 | 1.43(2.3) | 0.94 | 1.9(0.9) | 0.33 | 0.33(1) |
| CIFAR-100 | ResNet-50 | 17.72 | 2.81(2.2) | 5.54 | 2.17(1.1) | 1.7 | 1.7(1) |
| | ResNet-110 | 19.44 | 3.74(2.3) | 7.36 | 3.86(1.2) | 1.54 | 1.54(1) |
| | Wide-ResNet-26-10 | 14.99 | 3.11(2.1) | 2.69 | 2.69(1) | 2.02 | 2.02(1) |
| | DenseNet-121 | 19.82 | 2.58(2.3) | 2.24 | 2.34(1.1) | 1.27 | 1.27(1) |
| Tiny-ImageNet | ResNet-50 | 16.19 | 5.56(1.5) | 2.14 | 2.14(1) | 1.37 | 1.37(1) |
| ImageNet | ResNet-50 | 2.84 | 1.62(0.9) | 16.77 | 2.52(0.7) | 3.68 | 1.1(0.9) |
| 20 Newsgroups | Global-pool CNN | 18.39 | 5.14(4.1) | 9.03 | 1.48(1.6) | 2.06 | 2.06(1) |

Table 7: $ECE_{EW-sweep}(\%)$. Underline marks the lowest error among pre-temperature scaling values. Bold marks the overall lowest in the row. Optimal temperature is selected based on the lowest ECE on the validation set.

| Dataset | Model | Cross Entropy | | FLSD-53 | | AdaFocal | |
|---------|-------|------|------|------|------|------|------|
| | | Pre T | Post T | Pre T | Post T | Pre T | Post T |
| CIFAR-10 | ResNet-50 | 4.05 | 1.43(2.5) | 1.54 | 1.54(1) | 0.04 | 0.7(0.9) |
| | ResNet-110 | 4.38 | 1.34(2.7) | 1.83 | 1.32(1.1) | 0.4 | 0.4(1) |
| | Wide-ResNet-26-10 | 3.53 | 1.41(2.2) | 1.64 | 1.55(0.9) | 0.38 | 0.32(1.1) |
| | DenseNet-121 | 4.27 | 2.17(2.3) | 1.58 | 1.98(0.9) | 0.34 | 0.34(1) |
| CIFAR-100 | ResNet-50 | 17.72 | 0.51(2.2) | 5.51 | 2.36(1.1) | 1.89 | 1.89(1) |
| | ResNet-110 | 19.44 | 3.71(2.3) | 7.34 | 3.65(1.2) | 1.58 | 1.58(1) |
| | Wide-ResNet-26-10 | 14.92 | 2.62(2.1) | 2.62 | 2.62(1) | 2.25 | 2.25(1) |
| | DenseNet-121 | 19.82 | 3.12(2.3) | 2.25 | 2.31(1.1) | 1.47 | 1.47(1) |
| Tiny-ImageNet | ResNet-50 | 16.19 | 5.44(1.5) | 2.10 | 2.10(1) | 1.6 | 1.6(1) |
| ImageNet | ResNet-50 | 2.93 | 1.63(0.9) | 16.77 | 2.58(0.7) | 3.73 | 0.92(0.9) |
| 20 Newsgroups | Global-pool CNN | 18.38 | 5.53(4.1) | 8.95 | 2.13(1.6) | 2.22 | 2.22(1) |

Table 8: $\text{ECE}_{\text{EM-sweep}}(\%)$. Underline marks the lowest error among pre-temperature scaling values. Bold marks the overall lowest in the row. Optimal temperature is selected based on the lowest ECE on the validation set.

## H ECE ERROR BARS

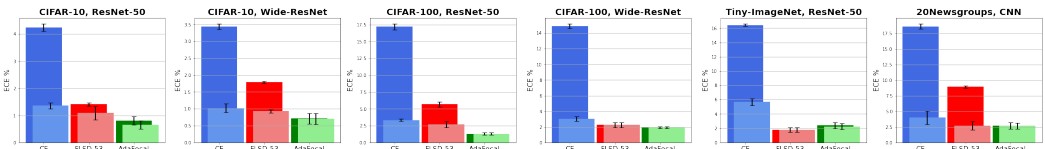

Figure 14: ECE error bars with mean and standard deviation computed over 5 runs with different initialization seed. The dark and light colors show pre and post temperature scaling results respectively.

## I NUMBER OF BINS USED DURING ADAFOCAL TRAINING

Experiment details: CIFAR-10, ResNet50 trained for 350 epochs. The reported results below are without temperature scaling. Our method AdaFocal with 5, 10, 15, 20, 30, and 50 adaptive (equal mass) bins vs FLSD-53. Note here that there are two types of binning:

- During training: the binning that is performed on the validation set from where AdaFocal draws calibration related information to adjust the $\gamma$. This corresponds to the columns in the table below.

- During evaluation: once we have a trained model, the binning that is used to compute different ECE metrics. These correspond to the rows in the table below.

|  | FLSD-53 | 5bins | 10bins | 15bins | 20bins | 30bins | 50bins |
|---|---|---|---|---|---|---|---|
| ECE(15bins) | 1.35 | 0.76 | 0.53 | 0.51 | 0.6 | 0.82 | 1.16 |
| AdaECE(15bins) | 1.67 | 0.63 | 0.53 | 0.56 | 0.4 | 0.84 | 1.1 |
| ECE_debias (15bins) | 1.62 | 0.5 | 0.44 | 0.47 | 0.25 | 0.79 | 1.07 |
| ECE_debias(30bins) | 1.57 | 0.73 | 0.43 | 0.46 | 0.27 | 0.72 | 1.06 |
| EW_ECE_sweep | 1.31 | 0.66 | 0.43 | 0.48 | 0.48 | 0.8 | 1.08 |
| EM_ECE_sweep | 1.54 | 0.53 | 0.21 | 0.04 | 0.38 | 0.07 | 1.08 |

Table 9: Effect of number of bins used for AdaFocal.

## J AUROC FOR OUT-OF-DISTRIBUTION DETECTION

For ResNet110 on CIFAR-10/SVHN, we were not able to reproduce the reported results of 96.74, 96.92 for FL-3 in Mukhoti et al. (2020). Instead we found those values to be 90.27, 90.39 and report them in Table 10

| Dataset | Model | Cross Entropy | | Brier Loss | | MMCE | | LS-0.05 | | FL-3 | | FLSD-53 | | AdaFocal |
|---|---|---|---|---|---|---|---|---|---|---|---|---|---|---|
|  |  | Pre T | Post T | Pre T | Post T | Pre T | Post T | Pre T | Post T | Pre T | Post T | Pre T | Post T | Pre T |
| CIFAR-10 / SVHN | ResNet-110 | 61.71 | 59.66 | 94.80 | 95.13 | 85.31 | 85.39 | 68.68 | 68.68 | 90.27 | 90.39 | 90.33 | 90.49 | **96.09** |
|  | Wide-ResNet-26-10 | 96.82 | 97.62 | 94.51 | 94.51 | 97.35 | 97.95 | 84.63 | 84.66 | 90.92 | 91.30 | 93.08 | 93.11 | **96.63** |
| CIFAR-10 / CIFAR-10-C | ResNet-110 | 77.53 | 75.16 | 84.09 | 83.86 | 71.96 | 70.02 | 72.17 | 72.18 | 80.11 | 79.78 | 82.06 | 81.38 | **84.96** |
|  | Wide-ResNet-26-10 | 81.06 | 80.68 | 85.03 | 85.03 | 82.17 | 81.72 | 71.10 | 71.16 | 83.33 | 84.00 | 80.00 | 80.76 | **89.52** |

Table 10: AUROC (%) of models trained on CIFAR-10 as the in-distribution data and tested on SVHN and CIFAR-10-C as out-of-distribution data.

## K MOVING AVERAGE $\gamma$-UPDATE RULE

For the focal loss in the paper $\mathcal{L}(p_n, t) = -(1 - p_n)^{\gamma_{b,t}} \log p_n$, the unconstrained $\gamma$-update rule for AdaFocal is given by

$$\gamma_{t+1} = \gamma_t * \exp(C_{val,t+1} - A_{val,t+1}) \tag{5}$$
$$= \gamma_t * \exp(E_{val,t+1}) \tag{6}$$

If instead we use exponential moving average to update $\gamma$, then the update rule (let's call it MA-$\alpha$) is given by

$$\gamma_{t+1} = (\gamma_t)^\alpha * \left(e^{E_{val,t+1}}\right)^{1-\alpha} \tag{7}$$

$$= \gamma_{t-1}^\alpha * e^{\alpha E_{val,t}} * e^{(1-\alpha)E_{val,t+1}} \tag{8}$$

$$= \gamma_{t-1}^\alpha * e^{[\alpha E_{val,t} + (1-\alpha)E_{val,t+1}]} \tag{9}$$

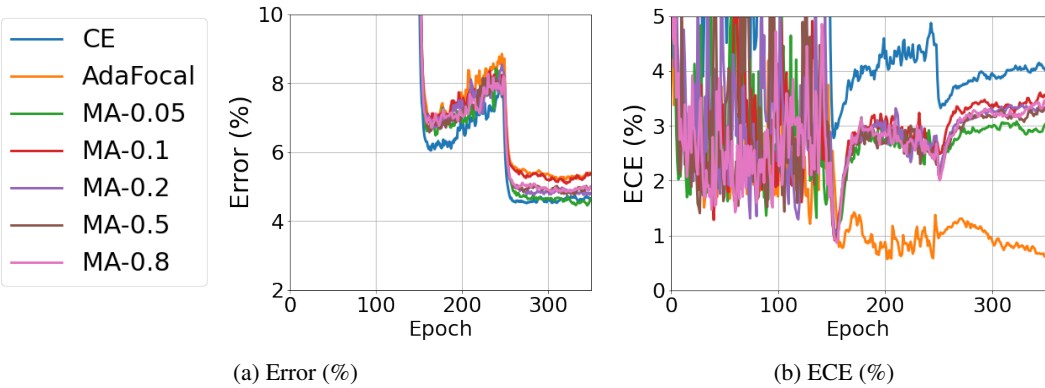

(a) Error (%)                       (b) ECE (%)

Figure 15: ResNet-50 trained on CIFAR-10 with cross entropy (CE), AdaFocal, and the moving average $\gamma$-update rule (MA-$\alpha$). We see that MA-$\alpha$ is not as effective as AdaFocal's $\gamma$ update rule.

The evolution or dynamics of $\gamma$ is given in Figure 16.

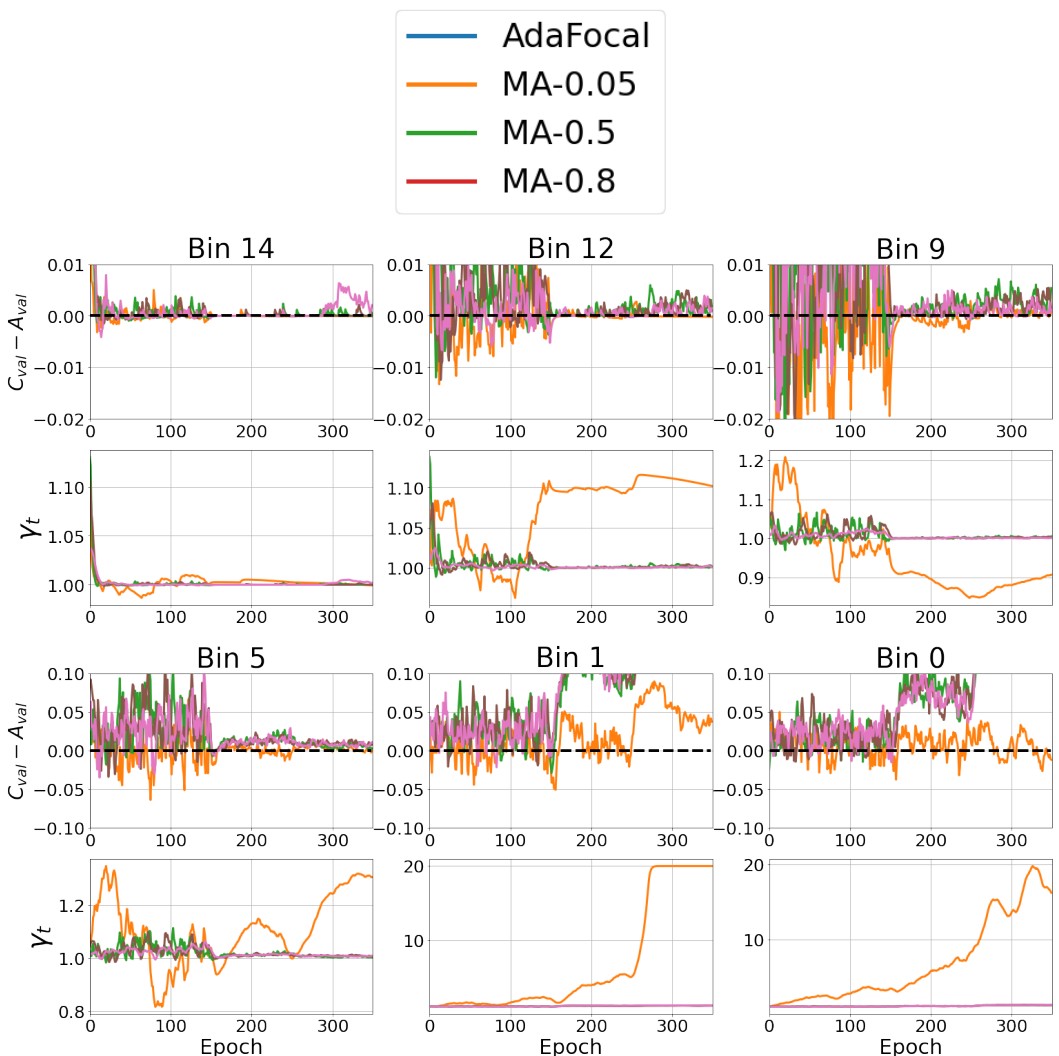

Figure 16: Training ResNet-50 on CIFAR-10 using AdaFOcal and MA-$\alpha$. Each bin has two subplots: **top**: $E_{val,i} = C_{val,i} - A_{val,i}$, **bottom**: evolution of $\gamma_t$. Black dotted line in top plot represent zero calibration error. We observe that for MA-$\alpha$ the $\gamma$ for different bins are not as free to move around as that under AdaFocal.

## L  MULTIPLE RUNS OF ADAFOCAL WITH DIFFERENT $\gamma_{\max}$

Due to the stochastic nature of the experiments, AdaFocal $\gamma$s may end up following different trajectories across different runs (initialization), which in turn might lead to variations in the final results. In this section, we look at the extent of such variations for (1) unconstrained $\gamma$ (2) $\gamma$ capped by $\gamma_{\max} = 20$ and (3) $\gamma$ capped by $\gamma_{\max} = 50$. We study this for ResNet-50 trained multiple times on CIFAR-10 starting with a different random seed.

### L.1  ADAFOCAL, $\gamma_{\max} = 20$

In Figure 17, we observe that AdaFocal with $\gamma_{\max} = 20$ is consistently (9 out of 9 times) better than FLSD-53. Figure 18 shows the evolution of $\gamma$ across different runs.

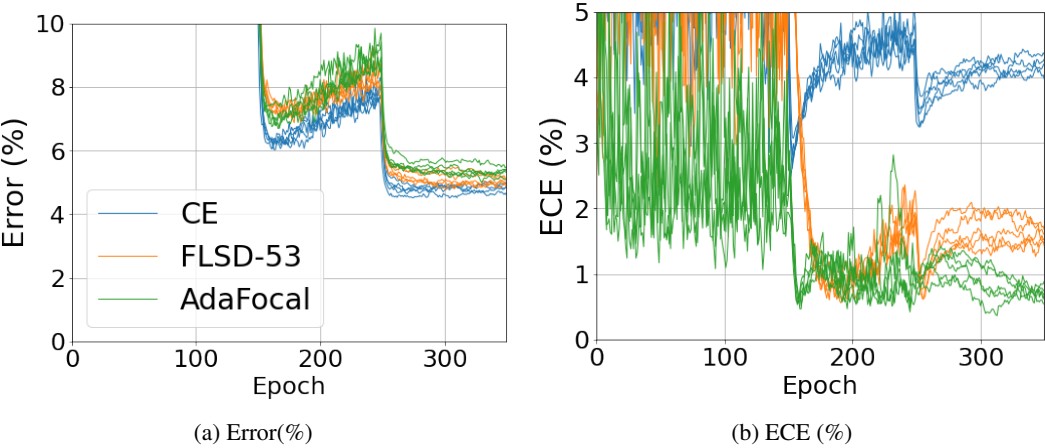

Figure 17: ResNet-50 trained on CIFAR-10 using 5 runs of cross entropy (CE), FLSD-53 and AdaFocal with different initialization seed. AdaFocal with $\gamma_{\max} = 20$ is consistently better.

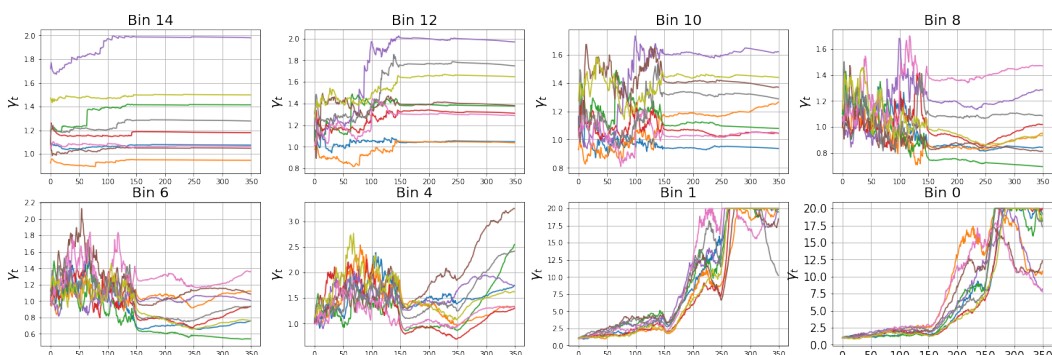

Figure 18: Evolution of $\gamma_t$ for 9 runs of ResNet-50 trained on CIFAR-10 using AdaFocal $\gamma_{\max} = 20$.

### L.2    ADAFOCAL, $\gamma_{\max} = 50$

In Figure 19, we observe that AdaFocal with $\gamma_{\max} = 50$ has more variability than AdaFocal $\gamma_{\max} = 20$ but is mostly better than FLSD-53. Figure 19 shows the evolution of $\gamma$ across different runs.

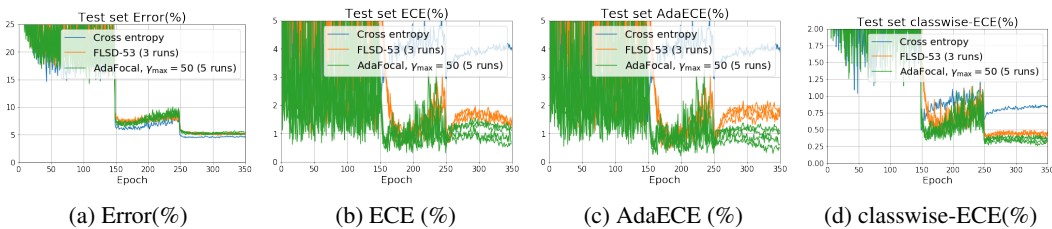

Figure 19: Plots for ResNet-50 trained on CIFAR-10 using cross entropy (1 run), FLSD-53 (3 runs) and AdaFocal (5 runs). AdaFocal with $\gamma_{\max} = 50$ although is mostly better than FLSD-53 it does exhibit greater variability than $\gamma_{\max} = 20$.

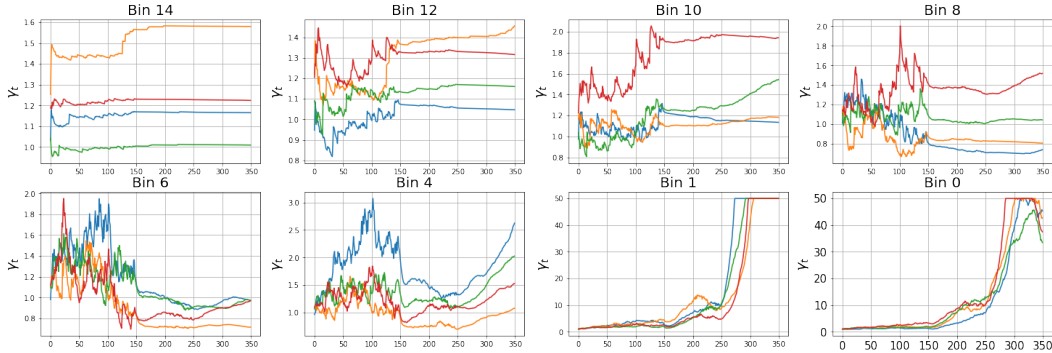

Figure 20: Evolution of $\gamma_t$ for multiple runs of ResNet-50 trained on CIFAR-10 using AdaFocal $\gamma_{\max} = 50$.

### L.3 AdaFocal, unconstrained $\gamma$

In Figure 21, we observe that AdaFocal with unconstrained $\gamma$ does exhibit some variability across different runs: 7 out of 9 times it performs better than FLSD-53 whereas the other two times it is similar or slightly worse.

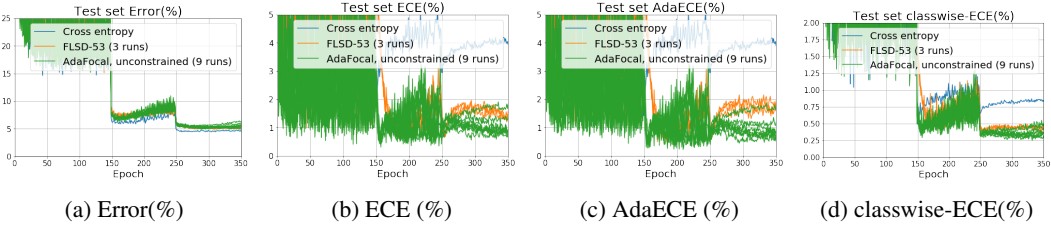

(a) Error(%)      (b) ECE (%)      (c) AdaECE (%)      (d) classwise-ECE(%)

Figure 21: Plots for ResNet-50 trained on CIFAR-10 using cross entropy (1 run), FLSD-53 (3 runs) and AdaFocal (9 runs). AdaFocal with unconstrained $\gamma$ does exhibit some variability across different runs: 7 out of 9 times it is better than FLSD-53, two times it is similar or slightly worse.

The above behaviour is mostly due to the variations in the trajectory of $\gamma$s for lower bins, as shown in Figure 22. For higher bins, we see the $\gamma$s settling to similar values, however for lower bins, as the $\gamma$s are unconstrained they blow up to very high values.

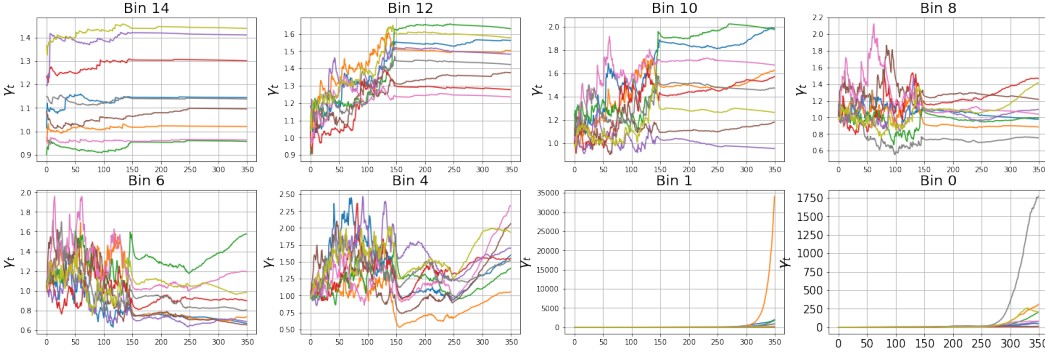

Figure 22: Evolution of $\gamma_t$ for multiple runs of ResNet-50 trained on CIFAR-10 using unconstrained AdaFocal.

# M   ERROR, ECE AND BIN STATISTICS PLOTS FOR REST OF THE EXPERIMENTS

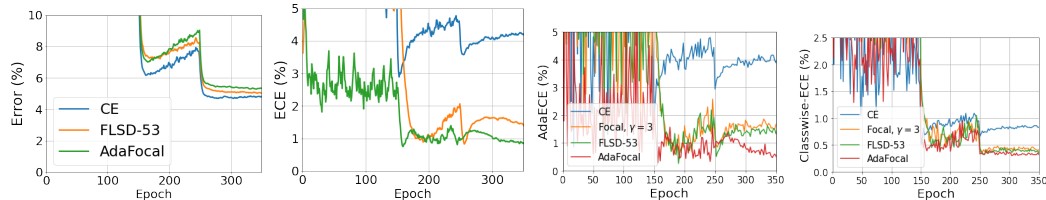

Figure 23: ResNet-50 trained on CIFAR-10 with cross entropy (CE), focal loss $\gamma = 3$, FLSD-53 and AdaFocal. (a) Error, (b) ECE, (c) AdaECE and (d) classwise-ECE. AdaFocal achieves the lowest calibration error while maintaining similar error performance.

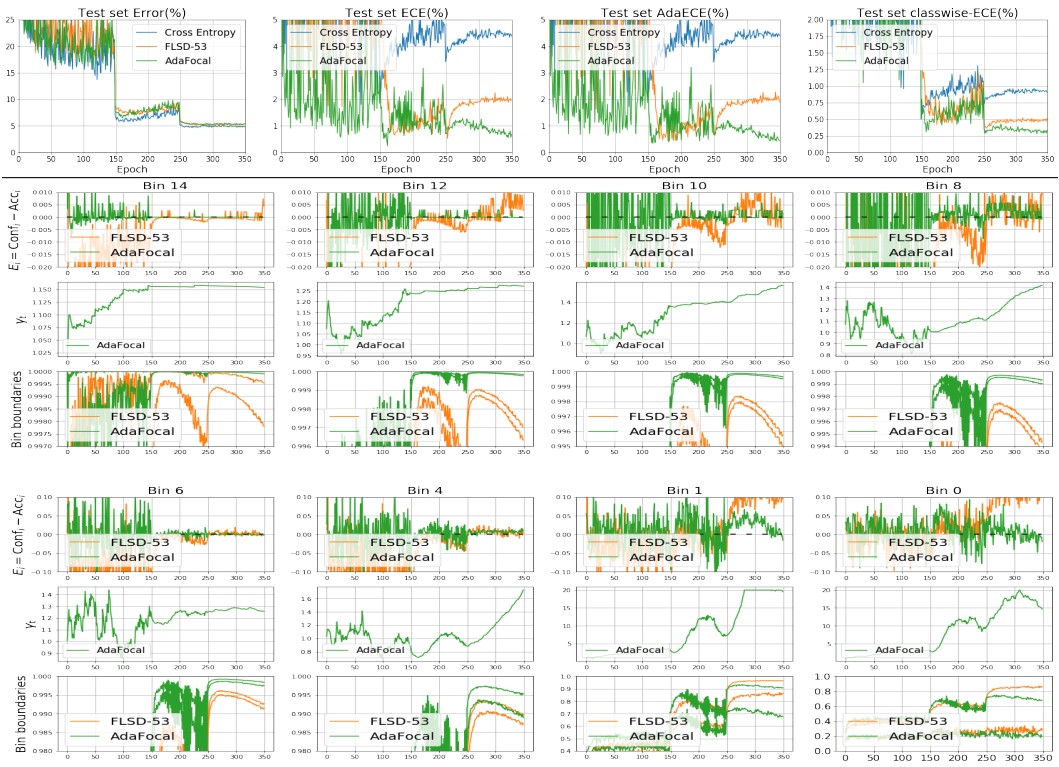

Figure 24: **CIFAR-10, ResNet-110**: Test set Error, ECE, AdaECE, classwise-ECE and Validation set bin information used by AdaFocal during training.

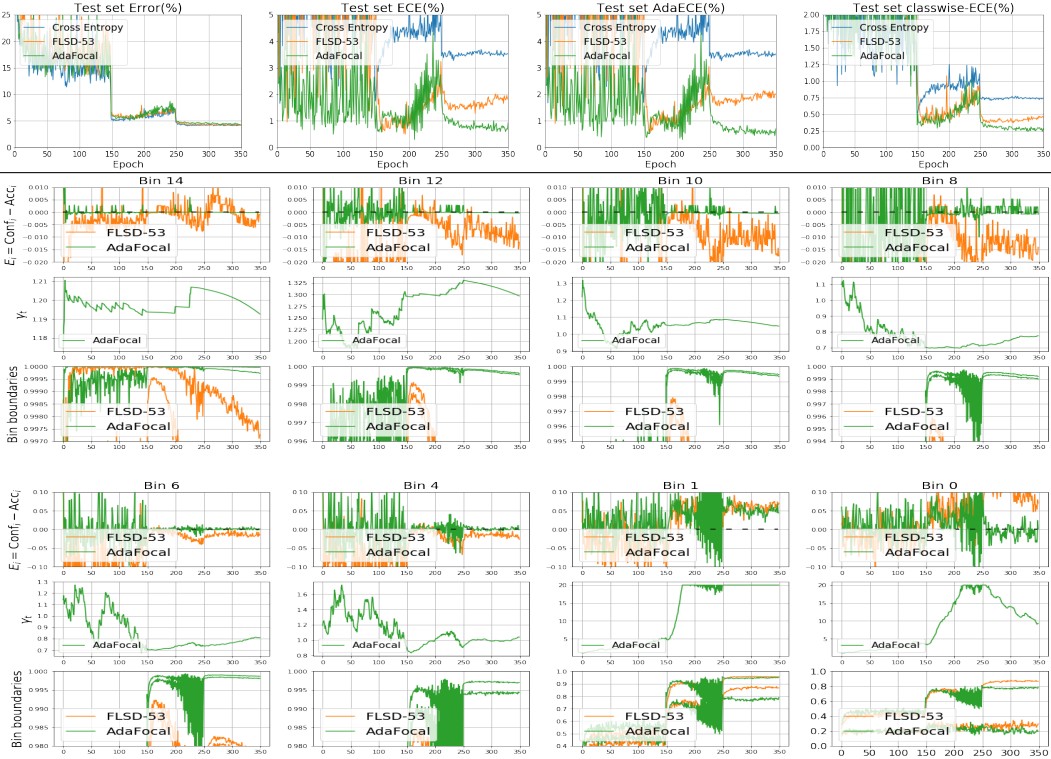

Figure 25: **CIFAR-10, Wide-ResNet**: Test set Error, ECE, AdaECE, classwise-ECE and Validation set bin information used by AdaFocal during training.

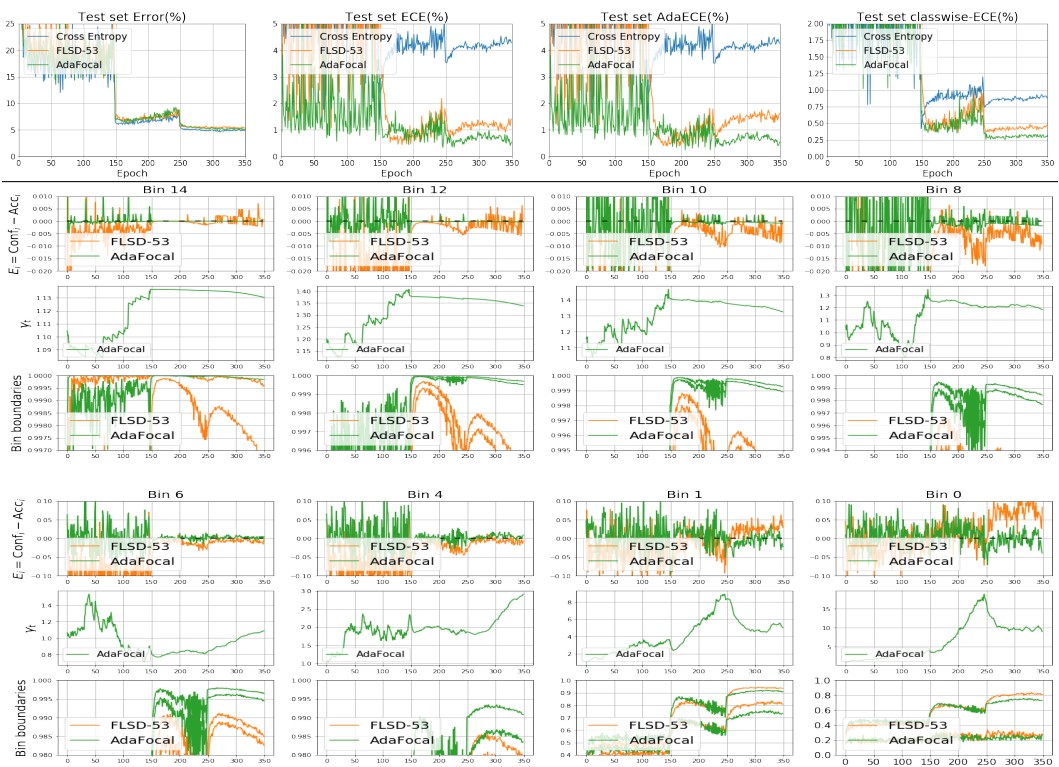

Figure 26: **CIFAR-10, DenseNet-121**: Test set Error, ECE, AdaECE, classwise-ECE and Validation set bin information used by AdaFocal during training.

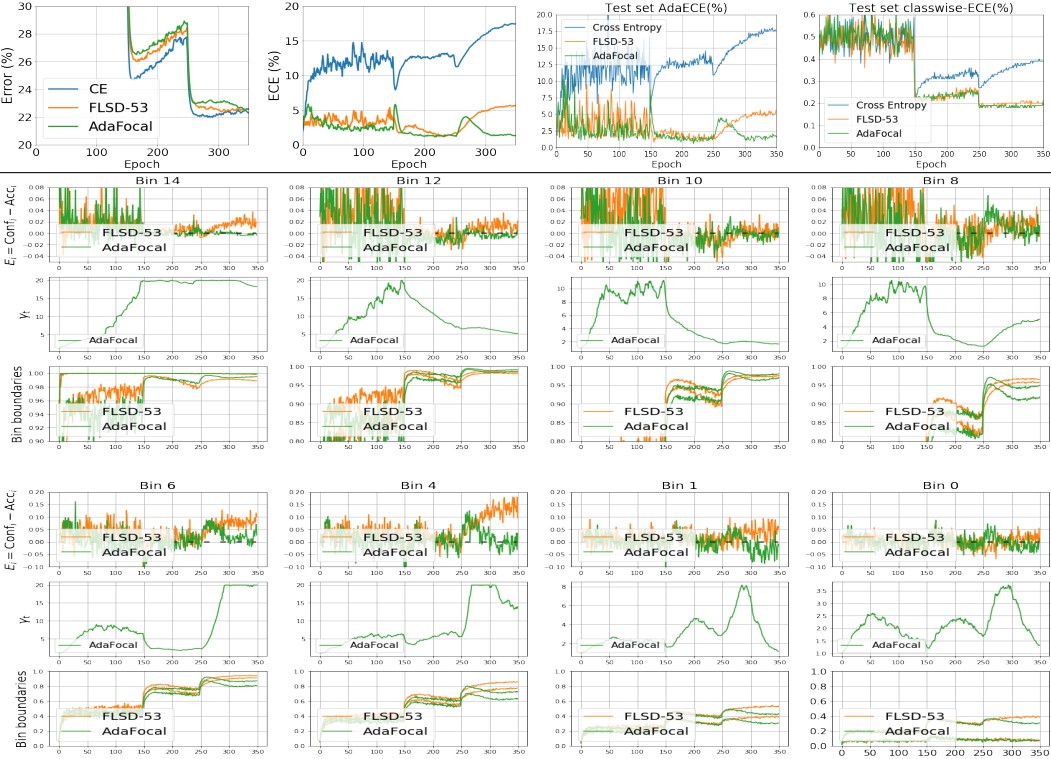

Figure 27: **CIFAR-100, ResNet-50**: Test set Error, ECE, AdaECE, classwise-ECE and Validation set bin information used by AdaFocal during training.

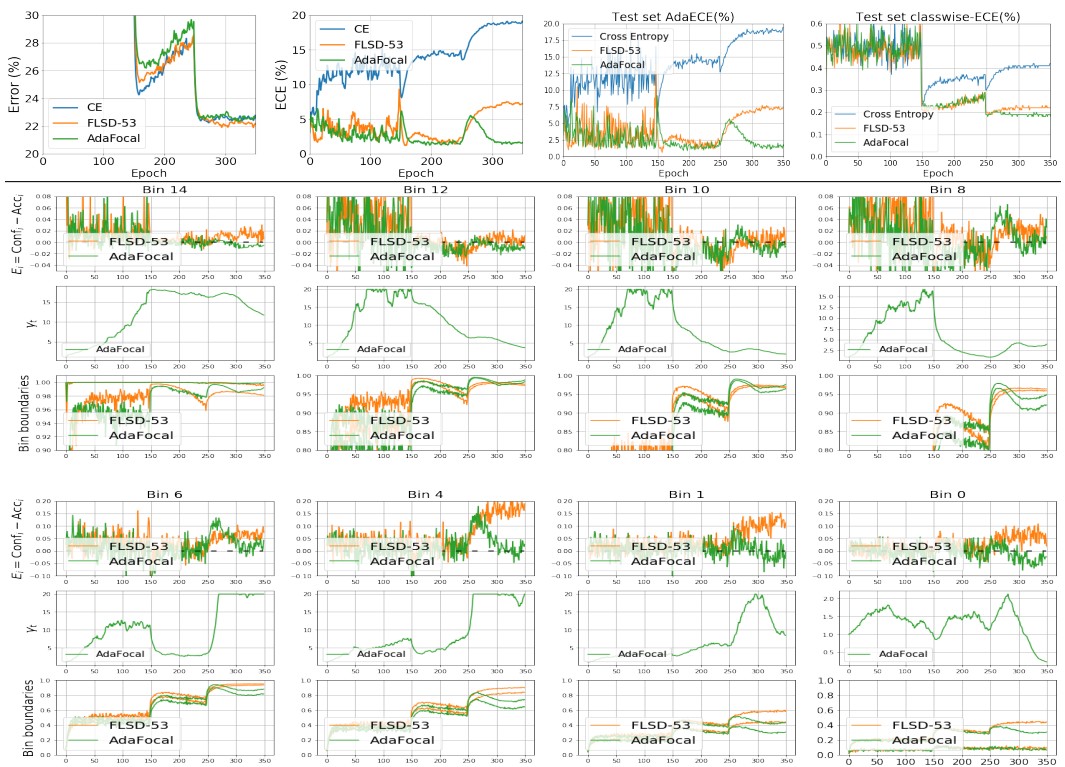

Figure 28: **CIFAR-100, ResNet-110**: Test set Error, ECE, AdaECE, classwise-ECE and Validation set bin information used by AdaFocal during training.

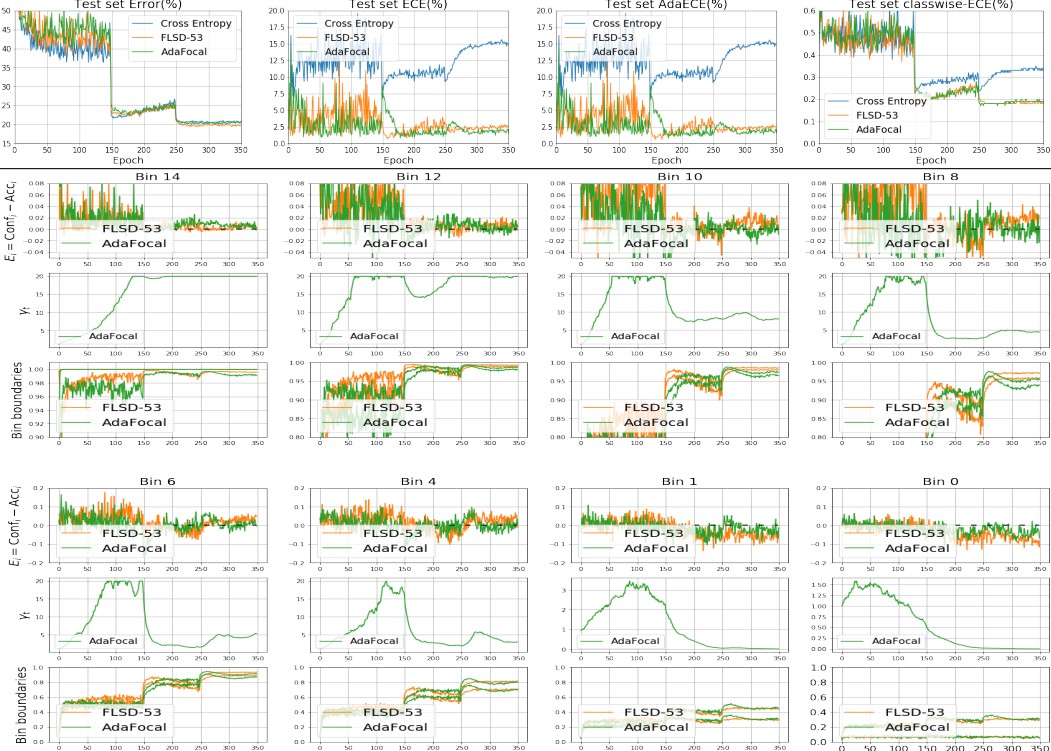

Figure 29: **CIFAR-100, Wide-ResNet**: Test set Error, ECE, AdaECE, classwise-ECE and Validation set bin information used by AdaFocal during training.

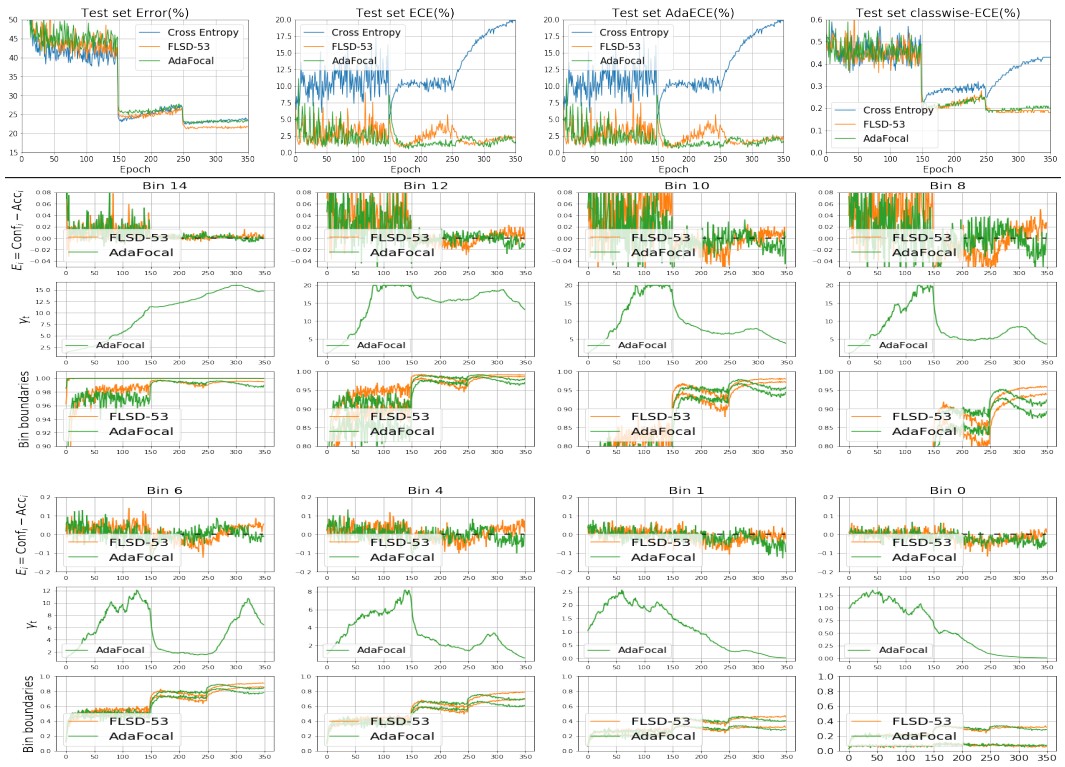

Figure 30: **CIFAR-100, DenseNet-121**: Test set Error, ECE, AdaECE, classwise-ECE and Validation set bin information used by AdaFocal during training.

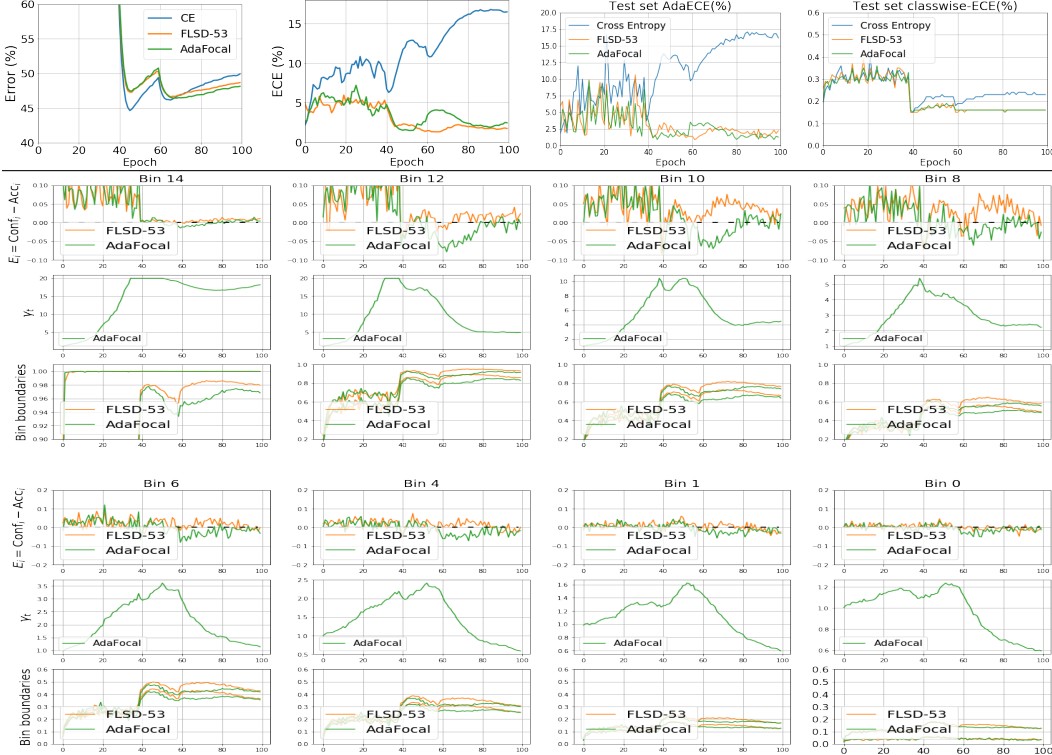

Figure 31: **Tiny-ImageNet, ResNet-50**: Test set Error, ECE, AdaECE, classwise-ECE and Validation set bin information used by AdaFocal during training.

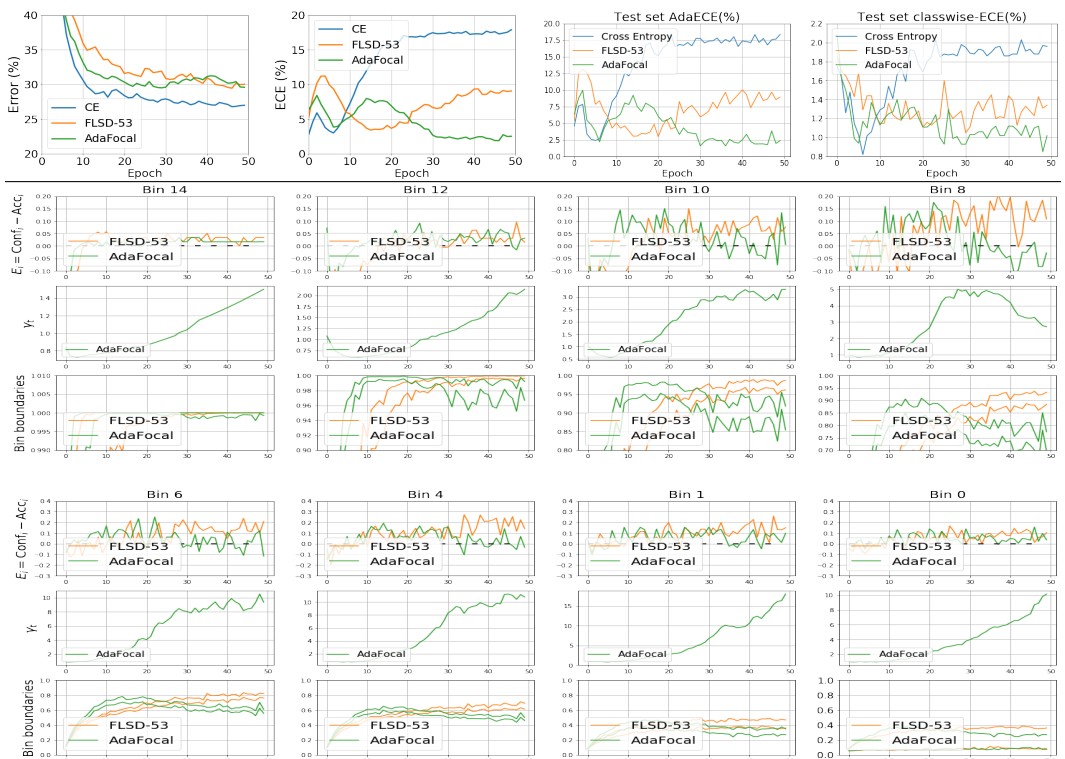

Figure 32: **20 Newsgroups, CNN**: Test set Error, ECE, AdaECE, classwise-ECE and Validation set bin information used by AdaFocal during training.

