# OpenReview forum: "AdaFocal: Calibration-aware Adaptive Focal Loss"
_ICLR.cc/2022/Conference — ICLR 2022 Submitted_

### Official Review · Reviewer_5y1y · 2021-10-30

**Correctness:** 4
**Technical Novelty And Significance:** 4
**Empirical Novelty And Significance:** 3
**Recommendation:** 6
**Confidence:** 3

**Main Review:**

# Pros

- The authors consider a wide variety of recent calibration literature.
- The approach is intuitive and is verified by empirical results showing that the patterns between validation and training ECE seem to hold.
- The authors motivate the need by empirically showing that a single focal loss parameter is not optimal among different bins.
- The experiments are extensive and cover important calibration and OOD metrics.

# Cons

- The axes and legends in figure 1 are too small and very hard to read
- The axes are unlabeled in figure 2a
- Section 5: “this mostly holds true” $\rightarrow$ It is not clear what “this” refers to. From the context it appears that “this” refers to “focal loss not leading to smaller gradients.” It appears the next sentence leads to the opposite conclusion which is confusing.
- Figure 5 is too cluttered and extremely hard to read
- Table 1 and the preceding paragraph: In both places, the text states that AdaFocal’s performance is averaged over 5 runs, which implies that the baselines are not. How many runs were the baselines trained for?
- The authors highlight that AdaFocal gives the best performance among the “Pre T” results, but isn’t it also the case that AdaFocal is the only algorithm which has made use of the validation data at this point? This would explain why there is not much improvement with temperature scaling and AdaFocal.
- The method of updating $\gamma$ is based on summing the exponents together which may lead to large values which are mitigated by clamping to a maximum value. Have the authors considered using an exponential moving average of the exponents such that the argument to the exponential is of the form $\alpha E_{val,t} + (1 - \alpha) E_{val,t+1}$? I believe this would prevent the instability of the sum and not require any clamping to a maximum value.



# Minor

- Page 3, section 4: The sentence which begins with “Nonetheless, for completeness…” is missing a prenthesis and probably also an unnecessary comma which makes the sentence confusing to read.
- Section 4: approaches towards $\rightarrow$ approaches.
- Section 5: During majority $\rightarrow$ during the majority
- Section 5.1: hyperparameter$\gamma$ $\rightarrow$ hyperparameter $\gamma$
- Section 5.2: this update rule do $\rightarrow$ this updates rules does
- Section 5.2: reign in $\rightarrow$ rein in


**Summary Of The Paper:**

The authors propose a new focal loss method of calibrating a model via regularization during training. They propose an adaptive method of adjusting the focal loss parameter by empirical validation performance which adjusts the parameter based on specific bin over/under confidence.

**Summary Of The Review:**

The authors do a good job in motivating their method and empirically showing why it is needed and why it should work. In mu opinion, the results are very hard to read and the paper has a problem with formatting/visibility or tables and figures. Additionally, If I have understood everything correctly, I think the formulation of equation 4 may be improved, eliminating a hyperparameter (see last bullet point in Cons section). Overall, these are the main points influencing my score.

---

> ### Author Response · Authors · 2021-11-22
> **Author response to Reviewer 5y1y [1/2]**
>
> We thank the reviewer for a thorough analysis of the paper and the review comments. They have been helpful to improve the paper. Please refer to our general comment "List of modifications in the latest revision" posted to all reviewers for the complete list of changes that we have made to the paper.
>
> > **Reviewer's comment:**
> > * The axes and legends in figure 1 are too small and very hard to read
> > * The axes are unlabeled in figure 2a
> > * Figure 5 is too cluttered and extremely hard to read
>
> **Author response:** We have fixed these issues. For changes to figures, please refer to the list of changes in the general comment to all reviewers.
>
> > **Reviewer's comment:** Section 5: “this mostly holds true” → It is not clear what “this” refers to. From the context it appears that “this” refers to “focal loss not leading to smaller gradients.” It appears the next sentence leads to the opposite conclusion which is confusing.
>
> **Author response:** Thank you for catching the mistake. We have fixed it in the modified draft. Our main message there was:
> 1. In the region p>0.2, increasing $\gamma$ leads to smaller gradients
> 2. In the region p<0.2, increasing $\gamma$ does not lead to smaller gradients (rather it leads to larger gradients)
> 3. Almost all the training samples lie in the region p>0.2
> 4. And as our goal is to reduce the gradients to stop over confidence, we simply stick to the rule of increasing $\gamma$ to decrease gradient (which is true for p>0.2) without worrying about what happens in the region p<0.2
>
> > **Reviewer's comment:** Table 1 and the preceding paragraph: In both places, the text states that AdaFocal’s performance is averaged over 5 runs, which implies that the baselines are not. How many runs were the baselines trained for?
>
> **Author response:** We do have the baseline results averaged over 5 or more runs as well. We have presented these multiple runs in the form of
> 1. Mean, variance bars in Figure 6 of the previous draft (in the latest draft, we have pushed this to Appendix H to free some space).
> 2. The Error and ECE plots for ResNet-50 trained over CIFAR-10, CIFAR-100 and Tiny-ImageNet presented in Figure 4 (in the latest draft) have been averaged over 5 runs.
> 3. Multiple runs of CE, FLSD-53 and AdaFocal plotted for each run in Appendix H in previous draft and Appendix L in the latest draft.
>
> For the results in Table 1 and Table 2, as mentioned in the footnote of page 8 (previous draft) and page 8 (latest draft), we have borrowed the exact values for the baseline results from the paper [Mukhoti et al. (2020)] to maintain consistency for comparison purposes. This is because for running the baseline experiments we have used the exact code from their official github repository (which includes the implementation of FLSD-53, MMCE, Brier Loss, Label Smoothing etc.) and found our baseline results to be very close to Table 1 and Table 2 reported in [Mukhoti et al. (2020)].
>
> Mukhoti et al. (2020): Jishnu Mukhoti, Viveka Kulharia, Amartya Sanyal, Stuart Golodetz, Philip HS Torr, and Puneet KDokania. Calibrating deep neural networks using focal loss. InAdvances in Neural InformationProcessing Systems, 2020.
>
> > **Reviewer's comment:** The authors highlight that AdaFocal gives the best performance among the “Pre T” results, but isn’t it also the case that AdaFocal is the only algorithm which has made use of the validation data at this point? This would explain why there is not much improvement with temperature scaling and AdaFocal.
>
> **Author response:** We agree with the reviewer that this comparison may seem unfair at first. However, note that the main goal of the paper was to compare the performance of AdaFocal in the space of “calibration during training itself” methods. While it may seem unfair that the other Pre-T (or “calibration during training”) methods do not get to use the validation set, this is a drawback because of their design itself. There is no way in their design to utilise a validation set during training. Further, it is not easy or immediately clear how one can utilise the validation set during training itself. AdaFocal is the first work to show such an algorithm.
>
> Another point to stress on is that if we already have a well calibrated model (from training) then post-hoc calibration on top may lead to further improvements (compared to post-hoc calibration on a miscalibrated model from training). Although simple temperature scaling on top of AdaFocal did not see much improvement, however, combining other state-of-the-art post-hoc calibration techniques may provide that gain. Exploring “AdaFocal + post-hoc calibration” is part of our future work.

---

> > ### Author Response · Authors · 2021-11-22
> > **Author response to Reviewer 5y1y [2/2]**
> >
> > > **Reviewer's comment:** The method of updating γ is based on summing the exponents together which may lead to large values which are mitigated by clamping to a maximum value. Have the authors considered using an exponential moving average of the exponents such that the argument to the exponential is of the form αEval,t+(1−α)Eval,t+1? I believe this would prevent the instability of the sum and not require any clamping to a maximum value.
> >
> > **Author response:** We thank the reviewer for the suggestion. Our main reason for choosing an exponential update was to achieve a rapid increase/decrease in $\gamma_t$ from the point $\gamma_{t-1}$. The doubt about moving average is that on one hand it may stop $\gamma_t$ from exploding, however on the other hand it may not lead to the desired rate of increase/decrease.
> >
> > We have run a few experiments, using the following moving-average update rule:
> > \begin{equation}
> >     \gamma_{t+1} = \left(\gamma_{t}\right)^{\alpha} * \left(e^{E_{val,t+1}}\right)^{1-\alpha}\\
> >                  = \gamma_{t-1}^{\alpha} * e^{\alpha E_{val,t}} * e^{(1-\alpha)E_{val,t+1}}\\
> >                  = \gamma_{t-1}^{\alpha} * e^{\left[\alpha E_{val,t} + (1-\alpha)E_{val,t+1}\right]}
> > \end{equation}
> >
> > for $\alpha=0.05, 0.1, 0.2, 0.5, 0.8$ to confirm the above points.
> >
> > The results are presented in Appendix K of the latest draft. We observe that:
> > 1. Moving average update is overall not as effective and the performance is quite far from AdaFocal (Figure 14 Appendix K).
> > 2. The reason being, as shown in Figure 15 Appendix K, the $\gamma$s don’t increase/decrease rapidly to the desired values. AdaFocal, on the other hand, is able to reach such high $\gamma$s.
> > 3. Moving average update also has a hyperparameter $\alpha$ which may require fine tuning. Whereas note that $\gamma_{max}$ of AdaFocal does not require special fine tuning and a reasonable upper bound in the range 20 to 30 works quite well.
> >
> > > **Reviewer's comment:** Minor: grammatical issues and typos
> >
> > **Author response:** Thank you for catching these mistakes. We have fixed them in the modified version.

---

> > > ### Comment · Reviewer_5y1y · 2021-11-25
> > > **Thanks for the clarifications**
> > >
> > > Thank you for taking the time to clarify the issues. On a second look through the paper, I have one more question relating to temperature scaling which I cannot find mentioned in the text...
> > >
> > > In every figure/table which does not explicitly refer to 'pre-T' and 'post-T', are any of the models temperature scaled?

---

> > > > ### Author Response · Authors · 2021-11-25
> > > > **Author response**
> > > >
> > > > No, a temperature scaled model is always referred by "post-T" added to its description. So, in the paper's main text, only Table 1 and Figure 6 have results for models after temperature scaling. Those are marked as Post-T. "AdaFocal" on its own refers to a model without temperature scaling for example in Figure 6.
> > > >
> > > > Since we cannot make any changes to the draft now, we would also like to take this opportunity to correct a few mistakes in our previous comment about the moving-average $\gamma$-update rule:
> > > > 1. One should refer to Figure 15 and Figure 16 in Appendix K (instead of Figure 14 and Figure 15 as per our previous comment) for experiments with moving-average update.
> > > > 2. In Fig. 16, I just realized the legends are incorrect. Correct legends for Fig. 16 should be the same as that in Fig. 15 i.e. Orange = AdaFocal, Green = MA-0.05, Brown = MA-0.5, Pink = MA-0.8.
> > > >
> > > > This does not change the conclusion of the experiments in any manner. But we hope that it did not create any confusion and that the message is still clear.

---

> > > > > ### Comment · Reviewer_5y1y · 2021-11-25
> > > > > **Update**
> > > > >
> > > > > Thanks for the clarification. I see now that everything that is evaluated on the test set has both metrics added, but the others are during training which is before it would be possible to apply temperature scaling.
> > > > >
> > > > > I do, however, strongly disagree with the idea that drawing a comparison to the 'pre-T' baselines in all the figures with temperature scaling is fair when temperature scaling is very easy to apply and AdaFocal has already made use of the information in the validation set. In my opinion, it would have been more appropriate to only show the 'post-T' numbers and show that the tuned temperature was 1 for AdaFocal, indicating that it was calibrated during training.
> > > > >
> > > > > Nonetheless , I think the authors have a good contribution here which does reasonably well at calibrating the model during training. The most compelling figure, in my opinion, is Figure 6 which shows a nice increase in OOD separability.
> > > > >
> > > > > I will update my score accordingly.

---

### Official Review · Reviewer_u2LZ · 2021-10-31

**Correctness:** 3
**Technical Novelty And Significance:** 3
**Empirical Novelty And Significance:** 3
**Recommendation:** 5
**Confidence:** 3

**Main Review:**

# Strong points

- The paper proposes an adaptive version of focal loss for model calibration. The hyper-parameters in focal loss can dynamically adjust for the calibration criterion.

- A similar validation set is used for updating the hyper-parameters. The updating rule is simple and can be easily plugged in existing methods.

- The proposed method outperforms other baselines on extensive datasets


# Weak points

- One glaring issue of the current version is the readability of figures. Fig (1),(2),(4),(5) are difficult to parse. There are even lines with duplicate colors in the second-row figures of Fig (2).b.

- "And from a calibration point of view, our strategy going forward would be to exploit this behavior to keep C_{train,true,i} (which we have control over during training) closer to A_{val,i} so that, in turn, C_{val,top,i} also stays closer to A_{val,i} to overall achieve low calibration error C_{val,top,i} − A_{val,i}". It seems to me there are two pitfalls in the sentence: (1) since the validation set is the actual target (claimed by the author on page 2), how can people make use of the accuracies on the validation set to tune models? (2) The correspondence in average confidence $C$ can not be directly translated into the correspondence in the calibration error $|C-A|$.

- It seems that the update rule in Eq. (4) is unstable. For example, if $C-A > k$ for $m$ steps, then $\gamma \ge e^{mk}$, which in turn makes the focal loss rather small. The authors propose to use a threshold to rein in the explosion. However, it seems to me that the explosion will commonly happen, and at most of the time $\gamma=\gamma_{max}$. Could the authors provide the dynamics of $\gamma$ during the training process?

# Minors

- In the second paragraph from the bottom: $\mathcal{L}_f$ should be the Focal loss L_Focal?

- It seems there is an extra "top" in the subtitle of Fig (1).b?

- Bottom paragraph on page 4: the $n$-th sample should be in $i$-th bin?

- Sentence above Eq (1): A_{val,b} instead of A_{val,i}

- What's the y-axis in Fig .(3)? Classification error on validation set?




**Summary Of The Paper:**

This paper considers the problem of model calibration. Existing works calibrates the model by post-hoc approaches or objective function tailored for calibration. The authors of the paper propose an adaptive version based on Focal loss, which regularizes the overconfidence of neural networks. They observe that although focal loss improves the calibration, it leaves out the under-confident samples. To mitigate the issue, they propose adjusting the hyper-parameter $\gamma$ in focal loss according to the model's under/over-confidence. Experiments on vision and NLP classification tasks showcase the effectiveness of the adaptive version.

**Summary Of The Review:**

All in all, the idea is interesting but I find the paper in its current form to be somewhat below the standard for ICLR acceptance, especially the write-up itself.

---

> ### Author Response · Authors · 2021-11-22
> **Author response to Reviewer u2LZ [1/2]**
>
> We thank the reviewer for a thorough analysis of the paper and the review comments. They have been helpful to improve the paper. Please refer to our general comment "List of modifications in the latest revision" posted to all reviewers for the complete list of changes that we have made to the paper.
>
> > **Reviewer's comment:** One glaring issue of the current version is the readability of figures. Fig (1),(2),(4),(5) are difficult to parse. There are even lines with duplicate colors in the second-row figures of Fig (2).b.
>
> **Author response:** For figures, please refer to the list of changes in the general comment to all reviewers.
>
> Regarding “duplicate colors” in the previous draft's figures, those were the bin boundaries (lower, upper] shown by the same color for a given loss function. For example, we had used orange for focal loss-3 where the two lines representing the lower boundary and upper boundary (both for focal loss-3) were marked orange. In the latest revision, we have moved the bin boundaries to the Appendix to save space as they were not central to the discussion in Fig 2.
>
> > **Reviewer's comment:** "And from a calibration point of view, our strategy going forward would be to exploit this behavior to keep C_{train,true,i} (which we have control over during training) closer to A_{val,i} so that, in turn, C_{val,top,i} also stays closer to A_{val,i} to overall achieve low calibration error C_{val,top,i} − A_{val,i}". It seems to me there are two pitfalls in the sentence: (1) since the validation set is the actual target (claimed by the author on page 2), how can people make use of the accuracies on the validation set to tune models? (2) The correspondence in average confidence C can not be directly translated into the correspondence in the calibration error |C−A|.
>
> **Author response:** If we understood reviewer’s question correctly, there are two misunderstandings here which we would like to clarify:
> 1. By “confidence of the validation set being target”, we meant that during training we can control the confidence on the training set using focal loss, however, the training set confidences are not the actual thing we care about. We care about how this “manipulation of confidence of training samples $C_{train,i}$ indirectly leads to the “manipulation of confidence of validation samples $C_{val,i}$ so that if we can obtain a well calibrated model on the validation set then we would expect this calibrated model to generalize to test set as well (which will the real target or goal).
> 2. In the paper, we have denoted the calibration error in each bin by $E_i = C_i - A_i$. For the validation set, we can write this with the subscript “val” as $E_{val,i} = C_{val,i} - A_{val,i}$. So in order to achieve a low calibration error on the validation set, we would want, for all bins i, $C_{val,i} - A_{val,i} = 0$ or in other word we would want  $C_{val,i}$  to stay close to $A_{val,i}$. We can’t do much about $A_{val,i}$ as that’s what we get out of the network based on its training. We can however push $C_{val,i}$ towards $A_{val,i}$ but we do not have a way to achieve this directly. So instead, relying on the correspondence between $C_{train,i}$ and $C_{val,i}$, if we try to keep $C_{train,i}$ closer to $A_{val,i}$, which in turn (based on the correspondence shown in the paper) will keep $C_{val,i}$ closer to $A_{val,i}$, thus reducing "val" set calibration error which is again $E_{val,i} = C_{val,i} - A_{val,i}$.
>
> > **Reviewer's comment:** It seems that the update rule in Eq. (4) is unstable. For example, if C−A>k for m steps, then γ≥emk, which in turn makes the focal loss rather small. The authors propose to use a threshold to rein in the explosion. However, it seems to me that the explosion will commonly happen, and at most of the time γ=γmax. Could the authors provide the dynamics of γ during the training process?
>
> **Author response:** Please refer to Figure 5 (the bottom subfigure under each bin) for the dynamics of $\gamma$. In the previous draft we had provided the $\gamma$ values for only one experiment i.e for CIFAR-10 Resnet-50. In the latest draft, please find the values $\gamma$ takes in different bins for an additional experiment of ImageNet ResNet-50.
>
> From these figures, we clearly see that $\gamma$ does not explode often. Rather they settle down to an appropriate value that leads to the best calibration. For example,
> 1. for CIFAR-10 ResNet-50, $\gamma$ = 1.3 (bin-14), 1.8(bin-12), 1.3(bin-10), 1.2(bin-8),1(bin-6), 2.2(bin-4), 15(bin-1), 20(bin-0).
> 2. For Imagenet Resnet 50 these $\gamma$ mostly → 0.
>
> The dynamics of $\gamma$ for the rest of the experiments (and for all bins) are given in Appendix M for reference of the reviewer.

---

> > ### Author Response · Authors · 2021-11-22
> > **Author response to Reviewer u2LZ [2/2]**
> >
> > > **Reviewer's comment:**
> > > * In the second paragraph from the bottom: Lf should be the Focal loss L_Focal?
> > > * It seems there is an extra "top" in the subtitle of Fig (1).b?
> > > *Sentence above Eq (1): A_{val,b} instead of A_{val,i}
> >
> > **Author response:** We have fixed these mistakes in the modified draft.
> >
> > > **Reviewer's comment:** Bottom paragraph on page 4: the n-th sample should be in i-th bin?
> >
> > **Author response:** Yes that’s correct, it was under the assumption that “if n-th sample falls into the i-th bin”. We have made this explicit in the modified draft.
> >
> > > **Reviewer's comment:** What's the y-axis in Fig .(3)? Classification error on validation set?
> >
> > **Author response:** Yes, that is on the validation set. (b) Error, (c) ECE for the validation set (d) average “train” and “val” confidence in validation-bin-0.

---

> > ### Comment · Reviewer_u2LZ · 2021-12-01
> > **The correspondence vs the absolute difference**
> >
> > I thank the authors for the response.
> >
> > Through the authors' response to the correspondence problem, I feel like the authors want to argue that the absolute difference $|C_{val,i}-C_{train,i}|$ is small, instead of the correspondence. I would suggest the author use the absolute difference in Fig.2. Further, it seems that the absolute differences are large in some cases in Fig.2.

---

> > > ### Author Response · Authors · 2021-12-03
> > > **Reasoning behind the use of "correspondence" instead of "absolute difference"**
> > >
> > > We thank the reviewer for raising the discussion as we would like to take this opportunity to clarify that in the paper we have restrained ourselves from talking about the difference $C_{train,i} - C_{val,i}$ (which would be a strong statement) and have focused only on the somewhat loose correspondence of  - if we decrease $C_{train,i}$ (as in Fig 2 by increasing $\gamma=$ 0 to 3 to 5) then we see that $C_{val,i}$ also decreases and vice versa. **We have not made any statement about by how much $C_{val,i}$ will increase/decrease if we increase/decrease $C_{train,i}$ by let’s say $x$**. The reason being we do not at present have a theoretical framework/understanding to quantify the exact relation between the two quantities $C_{train,i}$ and $C_{val,i}$. Further we see that, as also pointed out by the reviewer, the absolute difference is not consistent across different bins to make any sort of empirical statement as well. However, the good news is that for AdaFocal the existence of the correspondence is enough for it to be effective (as it has been specifically designed to utilize the correspondence) and therefore in this work we have not explored the exact relation between $C_{train,i}$ and $C_{val,i}$.
> > >
> > > **For a detailed explanation, please find below an example scenario to show why the “correspondence” is good enough for AdaFocal to work:**
> > > Let’s say at some point $t$ in training there's over confidence in bin-$i$ by $e$ amount i.e. $C_{val,i} - A_{val,i} = e$. Then, we would want $C_{val,i}$ to decrease by $e$. However, here we do not know by how much $C_{train,i}$ should be decreased so that $C_{val,i}$ decreases exactly by $e$ amount. AdaFocal tries to achieve this gradually over next epochs in a step by step manner i.e. at epoch $t+1$ we increase $\gamma$ to decrease $C_{train,i}$ by some amount (let’s say $x$) and check how much has $C_{val,i}$ decreased, and then proceed as per the following three cases:
> > > 1. $C_{val,i}$ is still $> A_{val,i}$. This means that decreasing $C_{train,i}$ by $x$ has, although, decreased $C_{val,i}$ but not by the desired amount of $e$. Then at epoch $t+2$ we would decrease $C_{train,i}$ further (by increasing $\gamma$ further) to additionally decrease $C_{val,i}$ as well.
> > > 2. $C_{val,i}$ is very close to $A_{val,i}$. This means that decreasing $C_{train,i}$ by $x$ has decreased $C_{val,i}$ almost exactly by $e$. Then, we are happy with the current value of $\gamma$ and we continue to train with it over the next epochs.
> > > 3. $C_{val,i}$ is now $< A_{val,i}$. This means that decreasing $C_{train,i}$ by $x$ has decreased $C_{val,i}$ by more than $e$ amount thus causing an overshoot and now there’s under-confidence. Then at epoch $t+2$ we would go the other way and increase $C_{train,i}$ so that $C_{val,i}$ is increased again to compensate for the overshoot. This way we continue with this oscillatory overshoot and undershoot over subsequent epochs until it finally settles down around a $\gamma$ which keeps $C_{val,i}$ very close to $A_{val,i}$.
> > >
> > > Therefore, to reiterate, AdaFocal does not care about $C_{train,i}$'s relation to $C_{val,i}$ or $A_{val.i}$. It just cares about how far $C_{val,i}$ is from $A_{val,i}$ and then uses $C_{train,i}$'s correspondence with $C_{val,i}$ to gradually reduce the gap between $C_{val,i}$ and $A_{val,i}$. Therefore for the proposed algorithm in the paper it is enough to show and rely on the correspondence. It would be great and insightful to say more about relations between the two quantities such as the absolute difference $|C_{train,i} - C_{val,i}|$, as suggested by the reviewer, however such a study is part of the future work.

---

### Official Review · Reviewer_veY9 · 2021-11-02

**Correctness:** 3
**Technical Novelty And Significance:** 3
**Empirical Novelty And Significance:** 2
**Recommendation:** 5
**Confidence:** 3

**Main Review:**

Strength:
1. This paper gives an in-depth analysis of the calibration problem in deep learning. The authors investigate the calibration properties of focal loss, observe that fixed $\gamma$ is not optimal for better calibration,  and find a correspondence to adjust $\gamma$ in each iteration adaptively.
2. The authors present many figures and investigation experiments for analysis and verification of effectiveness. The proposed method is effective in CIFAR10/100 and SVNH datasets.

Weakness:
1. The generalization of the proposed method is not promised. The observed problem of fixed $\gamma$ are empirical and may vary between datasets.  On the other hand, the optimal hyperparameters for the proposed method may also vary between datasets.  Besides,  it would be better to report the large-scale datasets' results, such as ImageNet. The current experimental results are not convincing.
2. The figures in this paper are not easy to understand. (a)There are too many figures, and hard to understand the representations of each line. Some legends even block important information. The authors may present less but important figures with concise conclusions in captions. (b) It would be better to smooth some lines in some figures. For example, figure 1 will be better if the lines are smoothed. (c)Some figures are too small to read after printing. And some figures are not readable without color printing.
3. How will the proposed method affect the test accuracies? Will the proposed method lead to an accuracy drop as the expense for improved calibration? Figure 4 reports the accuracies but is very hard to tell the accuracies difference.

**Summary Of The Paper:**

This paper studies the calibration of deep learning, which aims to make confidence store accurately describe predictions' correctness probabilities. The authors improve focal loss and propose a calibration-aware focal loss for better calibration.  The proposed approach adaptively adjusts the coefficient of focal loss according to the momentums and current predictions' confidence. The authors conduct experiments on SVNH, CIFAR10/100 datasets to verify the approach's efficacy.

**Summary Of The Review:**

The calibration problem is interesting and practical in real-world applications.  This paper presents an in-depth analysis of the calibration problem and proposes an effective method to improve it. However, the proposed method has not been proved to generalize well in different datasets. The authors may answer this question by a theoretical analysis or experiments in large-scale datasets. Another suggestion is to improve the figures of this paper. If the authors can address my questions, I am willing to raise my score.

---

> ### Author Response · Authors · 2021-11-22
> **Author response to Reviewer veY9 [1/2]**
>
> We thank the reviewer for a thorough analysis of the paper and the review comments. They have been helpful to improve the paper. Please refer to our general comment "List of modifications in the latest revision" posted to all reviewers for the complete list of changes that we have made to the paper.
>
> > **Reviewer's comment:** The generalization of the proposed method is not promised. The observed problem of fixed γ are empirical and may vary between datasets. On the other hand, the optimal hyperparameters for the proposed method may also vary between datasets. Besides, it would be better to report the large-scale datasets' results, such as ImageNet. The current experimental results are not convincing.
>
> **Author response:**
> We thank the reviewer for suggesting to run AdaFocal on ImageNet as we found some interesting results and good proof of AdaFocal’s generalization to large-scale datasets. However, given the limited time and resources, we were able to run only 3 experiments before the deadline of 22nd November 2021: cross entropy (CE), FLSD-53 and AdaFocal. We have included these results in the latest draft of the paper in:
> 1. Main paper: Figure 4(d)  Error and ECE plots;
> 2. Main paper: Figure 5(b)  - the dynamics/evolution of $\gamma$ during training for few bins
> 3. Main paper: Table 1 and Table 2
> 4. Appendix A which plots the dynamics/evolution of $\gamma$ for more bins.
>
> For Brier loss, MMCE and Label smoothing, we are still waiting for the training to finish therefore the respective columns in Table 1 and Table 2 are marked “-”. We can present the results as comments when available or present them in the final camera ready version if accepted.
>
> **Findings of the experiment:**
> For CIFAR-10, CIFAR-100 and Tiny-ImageNet, FLDS-53 is much better calibrated than CE. This is because, as shown in Figure 5(a), CE is over-confident compared to FLSD-53 in each bin. For ImageNet, however, the behaviour is reversed: FLSD-53 is poorly calibrated than CE. The reason, as shown in Figure 5(b), is that due to the use of high $\gamma$ values ($\gamma=5,3$), FLSD-53 makes the model largely under-confident in each bin, leading to an overall high calibration error.
>
> AdaFocal, on the other hand, is well calibrated for all the four dataset-model pairs. Particularly on ImageNet, calibration of AdaFocal is very close to that of CE. As per the dynamics/evolution of $\gamma_t$ in different bins as shown in Figure 5:
> (1) for CIFAR-10, they settle closer to $\gamma=1$ for higher bins and $\gamma=20$ for lower bins.
> (2) for ImageNet, AdaFocal's $\gamma \rightarrow 0$. This makes sense because as shown in Figure 4(d), cross entropy i.e. $\gamma=0$ for every bin is much better calibrated than FLSD-53. AdaFocal (starting from $\gamma=1$) finally settles down to CE ($\gamma=0$).
>
> This confirms that during training, unlike CE or FLSD-53, AdaFocal being aware of the network's current under/over-confidence (through the validation set) is able to adjusts the $\gamma$s in a way that maintains a well calibrated model at every step. Also note that for an unseen dataset-model pair it is difficult to know beforehand whether CE or focal loss will perform better. However, from these experiments, we find strong evidence that AdaFocal will eventually find the appropriate $\gamma$s that result in better calibration.
>
>
> **Possible reason why AdaFocal didn’t beat CE on ImageNet-ResNet50:**
> 1. As shown in Figure 5(b) “evolution of $\gamma_t$, the current $\gamma$-update rule takes a bit of time to bring down $\gamma_t$ from “1” (at the start of the training) to $\gamma \rightarrow 0$ towards the end of the training. Most likely a modification that can cause this decrease even more rapidly may lead to AdaFocal’s calibration being exactly the same as CE.
> 2. Note Bin-0 and Bin-6 in Figure 5(b), where even CE with $\gamma=0$ is under-confident i.e $C_{val}-A_{val} <0$. This implies that introduction of a loss function that can push the posterior even more aggressively $\rightarrow 1$ than CE may reduce the under-confidence in these lower bins to eventually beat CE on calibration as well.
>
> > **Reviewer's comment:** The figures in this paper are not easy to understand. (a)There are too many figures, and hard to understand the representations of each line. Some legends even block important information. The authors may present less but important figures with concise conclusions in captions. (b) It would be better to smooth some lines in some figures. For example, figure 1 will be better if the lines are smoothed. (c)Some figures are too small to read after printing. And some figures are not readable without color printing.
>
> **Author response:** For changes to figures, please refer to the list of changes in the general comment to all reviewers.

---

> > ### Author Response · Authors · 2021-11-22
> > **Author response to Reviewer veY9 [2/2]**
> >
> > > **Reviewer's comment:** How will the proposed method affect the test accuracies? Will the proposed method lead to an accuracy drop as the expense for improved calibration? Figure 4 reports the accuracies but is very hard to tell the accuracies difference.
> >
> > **Author response:** We have updated figure 4 with zoomed in view to the accuracy/error. We have also included Error and ECE plots from three other experiments to give a better view of the Accuracy and ECE performance.
> >
> > The slight drop or improvement in accuracy seems to be dependent on the dataset-model pair. Different architectures seem to behave differently across datasets, having different accuracies and can have different calibration performance. For example, from Figure 4 and Table 4 in main paper, we see that
> >
> > * No accuracy drop (AdaFocal compared to CE):
> >     1. CIFAR-100: ResNet50, ResNet-100, WideResNet, DenseNet
> >     2. Tiny-Imagenet: Resnet50
> >     3. ImageNet: Resnet50
> >
> > * Slight accuracy drop (AdaFocal compared to CE):
> >    1. CIFAR-10 ResNet-50, ResNet-110, DenseNet, WideResnet
> >     2. 20Newsgroup, CNN

---

### Official Review · Reviewer_3Y6N · 2021-11-10

**Correctness:** 4
**Technical Novelty And Significance:** 4
**Empirical Novelty And Significance:** 4
**Recommendation:** 6
**Confidence:** 4

**Main Review:**

The idea of the paper is to optimize \gamma tuning in focal loss to improve the calibration of neural networks model. The motivation makes a lot of sense. Although the novelty of the paper seems limited and idea seems adhoc, while practically the proposed approach could be helpful to the loss function and calibration study.  Some concerns to this paper are

[1] The paper seems written in a rush and the paper organization and writing is weak.
- In abstract, FLSD-53 is ambiguous to in its abbreviation when first used
- The plots in the paper mostly have tiny legends

[2] In table2, it seems AdaFocal  improved calibration but not accuracy in Cifar-10 data, while in the opposite way on Cifar-100 data, as compared to FLSD-53. Are there any interpretations about the such calibration and accuracy trade-offs?

**Summary Of The Paper:**

The paper propose a new sets of algorithms to modify \gamma in the focal loss, where a tunable \gamma is applied at different region of model predications. Empirically study on Cifar-10 and Cifar-100 data with various of benchmark networks show better ECE as compared to the other methods.

**Summary Of The Review:**

Overall the paper proposes a new strategy to optimize calibration via tuning \gamma in focal loss. Writing in this paper is a little concerning. The idea is a little incremental but could be potential beneficial to the community.

---

> ### Author Response · Authors · 2021-11-22
> **Author response to Reviewer 3Y6N**
>
> We thank the reviewer for a thorough analysis of the paper and the review comments. They have been helpful to improve the paper. Please refer to our general comment "List of modifications in the latest revision" posted to all reviewers for the complete list of changes that we have made to the paper.
>
> > **Reviewer's comment**: [1] The paper seems written in a rush and the paper organization and writing is weak.
> > * In abstract, FLSD-53 is ambiguous to in its abbreviation when first used
> > * The plots in the paper mostly have tiny legends
>
> **Author response**:
> 1. We have updated the abstract to include the full form of FLSD-53.
> 2. For figures, please refer to the list of changes in the general comment to all reviewers.
>
> > **Reviewer's comment**: [2] In table2, it seems AdaFocal improved calibration but not accuracy in Cifar-10 data, while in the opposite way on Cifar-100 data, as compared to FLSD-53. Are there any interpretations about the such calibration and accuracy trade-offs?
>
> **Author response**:
> The slight drop or improvement in accuracy seems to be dependent on the dataset-model pair. Different architectures seem to behave differently across datasets, having different accuracies and can have different calibration performance. For example, from Figure 4 and Table 4 in main paper, we see that
>
> * No accuracy drop (AdaFocal compared to CE):
>     1. CIFAR-100: ResNet50, ResNet-100, WideResNet, DenseNet
>     2. Tiny-Imagenet: Resnet50
>     3. ImageNet: Resnet50
>
> * Slight accuracy drop (AdaFocal compared to CE):
>     1. CIFAR-10: ResNet-50, ResNet-110, DenseNet, WideResnet
>     2. 20Newsgroup, CNN

---

### Author Response · Authors · 2021-11-22
**List of modifications in the latest revision**

1. We have added the experiments on ImageNet (a large-scale dataset) to the paper as per **Reviewer veY9’s** suggestion . However, given the limited time and resources, we were able to complete only 3 experiments before the deadline of 22nd November 2021: cross entropy (CE), FLSD-53 and AdaFocal. The rest of the experiments on MMCE Brier loss and Label-smoothing will not finish before the deadline and have been marked as “-” in the latest draft. We have included the ImageNet experiments in the following form
    1. Main paper: Figure 4(d) - Error and ECE plots
    2. Main paper: Figure 5(b)  - the dynamics/evolution of $\gamma$ during training for few bins
    3. Main paper: Table 1 and Table 2
    4. Appendix A which plots the dynamics/evolution of $\gamma$ for all bins.

Please refer to the paper or Author response to Reviewer veY9’s for a detailed discussion about the findings.

2. The plots in the paper have been smoothed using exponential smoothing with a smoothing-factor of 0.6.

3. Figure 1:
   1. Bins have been reduced from 0,1,6,19,14 to 0,7,14 to cover one bin each from lower, middle and upper region. The rest of the bins are shown in Appendix B.
    2. Subfigure size is increased. Axes and legends are now larger.

4. Figure 2:
    1. Bins have been reduced from 0,1,6,19,14 to 0,7,14.
    2. To avoid clutter, we have removed $A_{val}$ from the figure in the main paper as it is not the focus in the discussion of correspondence between $C_{train}$ and $C_{val}$. It was previously included to give a rough idea about how far $C_{train}$ and $C_{val}$ are from $A_{val}$.
    3. Have also removed the bin boundaries as it is not central to the discussion.
    4. Have switched the “independent binning” subfigure to top subfigure 2(a) and “common binning” subfigure to bottom subfigure 2(b)
    5. The legend has been changed for $C_{val}$ from “dashed-lines” to “starred-line” so that it is visible even if $C_{train}$ lies exactly on top of it.
    6. For reader’s reference rest of the bins with the original figure that includes $A_{val}$ and bin boundaries are now shown in Appendix C.

5. Figure 3:
    1. Have increased the figure, legend and axes size.
    2. The color in the legend of Fig 3(d) has been changed to match other subfigures.

6. Figure 4:
    1. Have included Error and ECE plots for ResNet-50 trained on cifar-100, tiny-imagenet, and imagenet to provide a better view of the accuracy and ECE behaviour across different dataset-model pairs.
    2. Have pushed the AdaECE and Classwise-ECE plots to Appendix M to free some space.
    3. Have removed the error and ECE curves for focal loss-3 to reduce clutter as its performance is always very close to FLSD-53. So the comparison of AdaFocal just against FLSD-53 is enough.

6. Figure 5:
    1. Have include the $C_{val}-A_{val}$ and dynamics/evolution of $\gamma_t$ plot for ImageNet as well
    2. Have reduced the number of bins and removed the bin boundaries to reduce clutter. For all bins and the bin boundaries refer to Appendix M.

7. Error bar plots in Figure 6 have been moved to Appendix H to free space.

8. AUROC Table 3 for OOD detection has been moved to Appendix J to free space. The improved performance of AdaFocal is clear from the ROC figure 6 itself in the main paper.

9. Have used $C_{val} \equiv C_{val,top}$ and $C_{train} \equiv C_{train,true}$ for clean notation and consistency throughout the paper.

10. Have pushed the discussion about “increase /decrease of $\gamma$ to decrease/increase focal loss gradient” (in the paragraph before Section 5.1 in previous draft) to footnote in the new draft to reduce space.

11. Have restructured section 5.1 with paragraph markings indicating which paragraph is for what.

12. In section 5.2, we have removed Eq. 3 in the previous draft and we directly start with Eq 4 of the previous draft (now Eq. 3 in the latest draft) as we never really used that equation in the final AdaFocal formulation.

13. Have pushed the discussion about “FLSD-53 performing very similarly to FL-3” to appendix as it is a comparison between these two particular focal losses and not central to the discussion about AdaFocal’s better performance compared to FLSD-53.

---

### Decision · Program_Chairs · 2022-01-20

**Decision:**

Reject

**Comment:**

The work AdaFocal proposes an approach to tune Focal Loss' $\gamma$ hyperparameter to improve the model's calibration, particularly to avoid the occasional underconfidence when using focal loss. This tuning is done not as a learned constant hyperparameter across training but as one that evolves over training.

The work is both well-motivated and well-written. However, multiple reviewers share the concern (which I agree with) that the method fails on ImageNet experiments, and the method often fails to beat even temperature scaling. The experimental comparison arguing that the approach improves upon many methods pre-temperature scaling is unfair as no other method leverages the validation set. This makes for a fairly deceiving slight of hand if not read carefully. When compared to temperature scaling, which does use the validation set, the performance improvement gap is diminished altogether.

I recommend the authors use the reviewers' feedback to enhance their preprint should they aim to submit to a later venue. In particular, improve the clarity around the experimental validation.